# HOIGS: Human-Object Interaction Gaussian Splatting from Monocular Videos

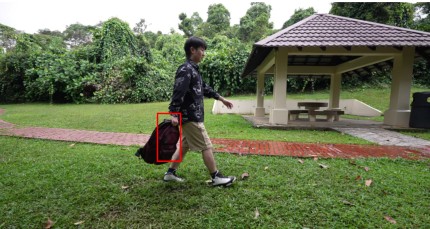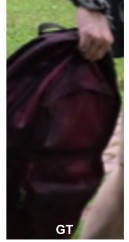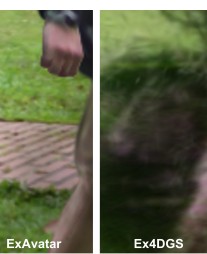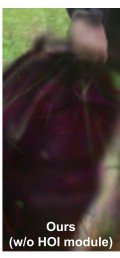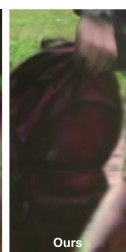

Figure 1: **Comparison between our method and previous approaches.** This figure compares rendering results between ExAvatar (Moon et al. (2024)), a human-centric model, and Ex4DGS (Lee et al. (2024)), which uses a single motion field for all motions. ExAvatar reconstructs only humans, while Ex4DGS fails to represent contact in interaction scenarios, producing artifacts and noise around contact regions.

## ABSTRACT

Reconstructing dynamic scenes with complex human–object interactions is a fundamental challenge in computer vision and graphics. Existing Gaussian Splatting methods either rely on human pose priors, neglecting dynamic objects, or approximate all motions within a single field, limiting their ability to capture interaction-rich dynamics. To address this gap, we propose Human-Object Interaction Gaussian Splatting (HOIGS), which explicitly models interaction-induced deformation between humans and objects through a cross-attention based HOI module. Distinct deformation baselines are employed to extract complementary motion features: hexplane for humans and Cubic Hermite Spline (CHS) for objects. By integrating these heterogeneous features, HOIGS effectively captures interdependent motions and improves deformation estimation in scenarios involving occlusion, contact, and object manipulation. Comprehensive experiments on multiple datasets demonstrate that our method consistently outperforms state-of-the-art human-centric and 4D Gaussian approaches, highlighting the importance of explicitly modeling human–object interactions for high-fidelity reconstruction. The video results of HOIGS are available at: `https://anonymous.4open.science/w/HOIGS-0F47/`

## 1 INTRODUCTION

Reconstructing videos of scenes that involve complex interactions between humans and objects and synthesizing novel viewpoints constitute a central research problem in computer vision and graphics. These techniques can be extended to various applications, including virtual reality, the metaverse, and 3D animation. However, the inherent limitations of monocular cameras and the need to accurately model intricate interactions between humans and objects remain major challenges for achieving high-quality reconstruction. Addressing these issues is essential for enabling realistic scene understanding and representation.

Recent approaches on human-centric video scene reconstruction (Kocabas et al. (2024); Moon et al. (2024); Hu et al. (2024c); Qian et al. (2024); Liu et al. (2024); Hu et al. (2024a); Wen et al. (2024); Kim et al. (2025)) have combined human pose estimation with 3D Gaussian Splatting (3DGS) (Kerbl et al. (2023)) to model dynamic scenes. Typically, SMPL (Loper et al. (2023)) parameters are

regressed in advance for each frame, and a canonical space is defined using a T-pose as the reference. Within this space, 3D Gaussian parameters are established and trained using feature planes and MLPs. Subsequently, deformation to each frame's 3D space is performed via Linear Blend Skinning (LBS) (Loper et al. (2023)), allowing for scene reconstruction and rendering. These methods have evolved into specialized models focused on humans and static backgrounds, achieving reliable performance when accurate human pose priors are available. However, existing approaches mainly focus on modeling humans alone, and thus fail to reconstruct complete scenes that involve objects beyond the human body. As a result, dynamically moving objects are often treated as static background or even disappear from the reconstructed scene. Even when deformations of objects are modeled separately, the interactions between humans and objects are not sufficiently considered in dynamic scenarios, which leads to artifacts and noisy results in the interaction regions, as shown in Fig. 1. Consequently, accurately reconstructing scenes that involve both humans and objects requires new modeling paradigms that extend beyond conventional human-centric frameworks.

Recent studies on 4D Gaussian Splatting extend beyond humans to encompass arbitrary moving objects, offering the advantage of general applicability. However, they generally exhibit lower reconstruction performance for humans compared to human-centric models. These approaches typically either define a canonical space and learn an implicit function that deforms it into the world coordinate system (Wu et al. (2024); Jung et al. (2023); Bae et al. (2024)), or explicitly parameterize object motions and optimize the corresponding parameters (Yang et al. (2023b); Li et al. (2024a); Lee et al. (2024)). Nevertheless, they do not explicitly model interactions between objects and instead treat all moving entities within a single motion field, which limits their ability to capture complex interactions. As a result, implicit methods struggle to represent long-term or highly non-linear motions in a stable manner, while explicit methods fail to handle scenarios such as contact and object manipulation, as ignoring the mutual interactions between motions limits their ability to capture realistic dynamics.

To overcome these limitations, we propose Human-Object Interaction Gaussian Splatting (HOIGS), a unified framework for reconstructing complex video scenes that involve both humans and dynamic objects. Unlike previous approaches that either model only human motion or employ a single motion field for all entities, our framework explicitly incorporates human–object interactions to achieve more faithful deformation modeling.

At the core of our framework lies the HOI module, which adopts a mutual attention mechanism to capture the bidirectional dependencies between human features and object motion features at each frame. Specifically, the module receives temporally varying human features, derived from the dynamic components of the hexplane representation, together with object motion features, obtained by embedding velocity vectors and their associated parameters. By explicitly learning how these two types of features influence one another, the HOI module effectively overcomes the shortcomings of prior methods that modeled humans and objects independently, which often resulted in artifacts and unstable reconstructions in interaction-rich scenes.

Furthermore, we design different deformation baselines tailored to humans and objects. For objects, we employ the Cubic Hermite Spline (CHS) to capture continuous motion trajectories, embedding the velocity vectors of keyframe Gaussians along with additional learnable parameters to construct robust object motion features. For humans, we utilize hexplane as the deformation baseline, where time-varying parameters are leveraged to represent fine-grained human deformation in both spatial and temporal domains. The extracted features from both humans and objects are subsequently integrated within the HOI module, which outputs offset vectors for each entity. This design ultimately enables our framework to achieve accurate and stable deformation estimation, even under complex scenarios involving close contact, mutual manipulation, or other intricate human–object interactions.

In summary, our main contributions are as follows:

- We propose an entity-aware cross-attention HOI module that enforces motion consistency between humans and objects. By attending to their features, it captures interdependent dynamics and improves reconstruction during contact and manipulation.

- We design distinct strategies for humans and objects using tailored deformation baselines. Hexplane encodes temporal and spatial features for human motion, while Cubic Hermite Splines (CHS) embed velocity vectors and learnable parameters for objects. This separation enables accurate and expressive motion representations for both entities.

- We conduct extensive experiments on diverse human–object interaction scenes and demonstrate that our method achieves more accurate reconstruction compared to existing human-centric and 4D Gaussian approaches.

## 2 RELATED WORKS

### 2.1 HUMAN MODELING

Research on realistic human modeling has long been pursued. Early parametric models enabled efficient estimation of human pose, exemplified by HMR (Kanazawa et al. (2018)), but struggled to capture clothing and accessories. To address this, implicit function-based methods (Huang et al. (2020); Saito et al. (2019; 2020); Xiu et al. (2022; 2023)) were proposed, which recover fine details such as hair and clothing but remain limited in global consistency and rendering efficiency. These methods mainly focused on human geometry with little attention to human-object interactions. With Neural Radiance Fields (NeRF) (Mildenhall et al. (2021)), several works applied it to human modeling (Peng et al. (2021); Jiang et al. (2022); Weng et al. (2022); Alldieck et al. (2022); Liao et al. (2023); Guo et al. (2023)), achieving realistic appearance and view consistency but still suffering from high training cost and slow rendering. In terms of human-object interactions, some attempts (Fan et al. (2024)) introduced objects, yet dynamic interactions were not fully captured. Recently, 3D Gaussian Splatting (3DGS) (Kerbl et al. (2023)) emerged as a new representation and has been applied to human reconstruction (Kocabas et al. (2024); Moon et al. (2024); Hu et al. (2024c); Liu et al. (2024); Hu et al. (2024a)). However, most efforts still regard objects as static. Recent approaches leverage multi-view data for high-fidelity results. For example, Animatable Gaussian Li et al. (2024b) and GASPACHO Mir et al. (2025) utilize multi-view setups to create relightable avatars and disentangle human-object interactions. However, these methods depend on complex capture settings, whereas real-world scenarios are predominantly monocular. To address this, we propose HOIGS, a model for stable human reconstruction that explicitly captures human–object interactions from monocular inputs.

### 2.2 DYNAMIC SCENE MODELING

The field of dynamic scene rendering and reconstruction has seen a paradigm shift from initial NeRF-based methods (Park et al. (2021a;b); Wu et al. (2022); Fridovich-Keil et al. (2023)) to the more recent 3D Gaussian Splatting framework. Previous studies such as HOSNeRF (Liu et al. (2023)) effectively modeled human-object interactions by controlling human motion through skeleton-based models such as SMPL and leveraging object state embeddings. Nevertheless, the implicit representation inherent to NeRF led to significant computational overhead in training and rendering, and limited the ability to represent detailed features in large-scale environments. To address this efficiency bottleneck, a line of work has emerged that extends 3DGS to the temporal domain, known as 4D Gaussian Splatting (4DGS) (Wu et al. (2024); Yang et al. (2023b)). Although these methods achieve real-time rendering speeds, they face persistent issues. Most 4DGS approaches rely on Structure-from-Motion for Gaussian initialization, which is fundamentally ill-suited for dynamic subjects as it operates on the assumption of a static world. This leads to inaccurate point cloud generation for moving objects. Moreover, the MLP-based implicit deformation fields used to capture motion, while adequate for simple trajectories, often result in over-smoothed or unnatural movements when applied to complex, in-the-wild scenarios. Therefore, we propose an explicit, spline-based motion model. This approach allows us to model intricate temporal movements with high fidelity, achieving high-quality rendering even in dynamic scenes that include complex human-object interactions.

## 3 METHOD

As shown in Fig. 2, we reconstruct the scene by independently modeling the deformations of humans and objects, and then incorporating interaction-aware transformations through the HOI module. Object deformations are estimated using a Cubic Hermite Spline (CHS). Human deformations are based on hexplane features, where time-invariant spatial features are used to learn the texture of the canonical T-pose, and Linear Blend Skinning (LBS) is subsequently applied to deform the canonical representation into each world space. Using these deformation baselines, we independently model

Figure 2: **Overview of the Proposed Framework.** Given an input video sequence, we first extract object-specific information, which is then used to reconstruct the 3D object shape via a diffusion prior. Based on the reconstructed shape, we initialize 3D Gaussians for each keyframe and use spline-based deformation as the baseline, where time-invariant and time-varying hexplane features are employed for canonical humans and interaction modeling, respectively. The final deformation is modeled through the HOI module, which learns interactions using human features and object motion features.

humans and objects and estimate their approximate positions for each frame, from which motion features are extracted. Finally, the extracted human and object features are fed into the HOI module, which accounts for interaction-driven transformations and determines the final positions of humans and objects in the reconstructed interaction scene.

## 3.1 OBJECT DEFORMATION

**Object Initialization.** We begin by segmenting the object of interest and cropping the object region from a representative frame of the entire sequence. Next, we apply a diffusion model initialized from DreamScene4D Chu et al. (2024) and guided by SDS loss to generate a canonical 3D Gaussian point cloud of the object. However, the 3D Gaussians generated through this diffusion prior may differ from the actual object geometry. While diffusion models can generate plausible 3D shapes from images, they often fail to precisely recover the true object structure. To address this, we align the diffusion-based canonical 3D Gaussians with the world space. First, we estimate a per-frame warping scale $S_t$ by minimizing the discrepancy between the projected 3D bounding box and the 2D mask bounding box for each frame $t$. We then compute the global scale $S$ as the average over all frames:

$$S_t = \arg\min_{S_t} \left\| \text{BBox}_{\text{proj}}(S_t \cdot \text{Gaussians}) - \text{BBox}_{\text{mask}}^t \right\|, \quad S = \frac{1}{T} \sum_{t=1}^{T} S_t.$$

where $T$ is the total number of frames. We then transform the scaled Gaussians to the world coordinate system using the COLMAP camera extrinsics $R_t$ and $\mathbf{t}_t$ as follows:

$$\mathbf{x}_{\text{world}}^t = R_t^{-1}(\mathbf{x}_{\text{cam}}^t - \mathbf{t}_t).$$

From the warped Gaussians $G_k$ of each keyframe, we extract each Gaussian's mean and color value, while initializing the remaining 3D Gaussian parameters with identity values.

**Object Deformation.** Based on the redefined mean and color from the keyframes, we construct the object's 3D Gaussians and use them to model the object deformation. To represent the continuous motion of the object over time, we model the mean values of each Gaussian as control-point-based curves. Specifically, we define a Cubic Hermite Spline function $\text{CHS}(t, \mathbf{m})$, and estimate the position of an object Gaussian at time $t$, denoted as $\mathbf{M}(t)$, as follows:

This initialization ensures that the explicit 3D Gaussian deformation model aligns with the actual object geometry and structural information. From the warped Gaussians of each keyframe, we extract

each Gaussian's mean and color value, while initializing the remaining 3D Gaussian parameters with identity values. We apply a diffusion prior with SDS loss to reconstruct the object from a representative frame of the entire sequence. The reconstructed object is then warped using the camera parameters of each keyframe to initialize the corresponding 3D Gaussians. However, the 3D Gaussians generated through the diffusion prior may differ from the actual object geometry. While diffusion models can generate plausible 3D shapes from images, they often fail to precisely recover the true object structure. To address this, we introduce an explicit 3D Gaussian deformation model that aligns the diffusion-based initialization with the actual object geometry and structural information. From the warped Gaussians $G_k$ of each keyframe, we extract each Gaussian's mean and color value, while initializing the remaining 3D Gaussian parameters with identity values. Based on the redefined mean and color from the keyframes, we construct the object's 3D Gaussians and use them to model the object deformation. To represent the continuous motion of the object over time, we model the mean values of each Gaussian as control-point-based curves. Specifically, we define a Cubic Hermite Spline function $CHS(t, \mathbf{m})$, and estimate the position of an object Gaussian at time $t$, denoted as $M(t)$, as follows:

$$M(t) = CHS(t, \mathbf{m}), \tag{1}$$

where $\mathbf{m} = \left\{ m_k \mid m_k \in \mathbb{R}^3 \right\}_{k \in [0, N_{key}-1]}$ is a learnable set of control points representing the mean positions of the Gaussians at each key frame, and $N_{key}$ denotes the number of key frames. $CHS(t, \mathbf{m})$ is formulated as

$$
\begin{aligned}
CHS(t, \mathbf{m}) = (2t_r^3 - 3t_r^2 + 1)m_{\lfloor t_s \rfloor} + (t_r^3 - 2t_r^2 + t_r)\tau_{\lfloor t_s \rfloor} \\
+ (-2t_r^3 + 3t_r^2)m_{\lfloor t_s \rfloor + 1} + (t_r^3 - t_r^2)\tau_{\lfloor t_s \rfloor + 1},
\end{aligned}
\tag{2}
$$

where $t_r = t_s - \lfloor t_s \rfloor$, $t_s = t_n(N_{key} - 1)$, $t_n = \frac{t}{N_f - 1}$ and $N_f$ denotes the number of all frames. $m_{\lfloor t_s \rfloor}$ denotes the mean of the 3D Gaussians corresponding to the $\lfloor t_s \rfloor$-th key frame.

In the standard formulation, $\tau_{\lfloor t_s \rfloor}$ represents the tangent vector with respect to the means of the surrounding Gaussians, which is typically approximated as $\tau_{\lfloor t_s \rfloor} = \frac{1}{2} \left( m_{\lfloor t_s \rfloor + 1} - m_{\lfloor t_s \rfloor - 1} \right)$. Instead of using this fixed approximation, we reinterpret $\tau_{\lfloor t_s \rfloor}$ as a *velocity vector* and employ it as a learnable parameter. By embedding this velocity, we construct motion features that better capture the dynamic behavior of objects over time.

The position parameter $\mathbf{m}$ between key frames is estimated via spline interpolation using both the Gaussian positions $m_k$ at the key frames and the corresponding velocity vectors $\tau_{\lfloor k \rfloor}$. Only the Gaussians at the key frames are directly optimized during training. Once the intermediate Gaussians are estimated and rendered, the resulting gradients from the loss function are backpropagated to update the parameters of the corresponding key frame Gaussians. Among the Gaussian parameters, rotation and opacity are defined as time-dependent variables. The rotation parameter is modeled using Spherical Linear Interpolation based on the Gaussian rotations at each key frame, enabling smooth transitions over time. The opacity parameter varies with time to account for occluded regions caused by object motion. In contrast, the scale parameter is kept constant across all corresponding Gaussians at different key frames.

## 3.2 HUMAN DEFORMATION

**Hexplane-based Deformation.** We model human deformation using hexplane features. Specifically, we adopt time-invariant spatial features $f$ from hexplane to learn the texture of the canonical T-pose mesh $T_c$ in the canonical space. The features $f$ are processed by an MLP head $\psi$ to learn the Gaussian properties in the canonical space. This representation serves as the baseline for human deformation. The canonical human representation is then deformed into the posed world space using Linear Blend Skinning (LBS) as follows:

$$\psi_h(f(T_c)) = (c, o, \Delta P_c, R, S, W), \tag{3}$$

$$P_{def} = \alpha * LBS(P_c, \theta, W), \tag{4}$$

where $\theta$ denotes the set of SMPL-X pose parameters and $\alpha$ is a learnable scale parameter for human pose. Equation (3) extracts the Gaussian properties (color $c$, opacity $o$, position offset $\Delta P_c$, rotation $R$, scale $S$ and skinning weights $W$) from the canonical hexplane features, while Equation (4) applies the LBS function to obtain the deformed positions $P_{def}$ of the Gaussians in the posed space.

To ensure that the reconstructed human representation matches the actual geometry, we further apply a depth supervision loss:

$$\mathcal{L}_{depth} = \|D_{render} - D\|_1 \,, \tag{5}$$

where $D_{render}$ is the rendered depth map from the deformed Gaussians and $D$ is the depth obtained from an off-the-shelf metric depth estimation model and further scaled using the COLMAP point cloud. This depth-guided supervision constrains the learnable scale parameter $\alpha$ and improves geometric fidelity in the reconstructed human shape.

### 3.3 CROSS-ATTENTION INTERACTION MODULE

**Feature Extraction.** We extract time-varying features from both humans and objects to learn their interactions. For humans, instead of relying on time-invariant texture features from the canonical space, we utilize time-varying features from hexplane. Furthermore, since it is not possible to know in advance which body parts are involved in object interactions, we divide the human body into 16 parts and extract hexplane features for each part.

For objects, the features are derived from the velocity embeddings associated with each keyframe in the deformation process, which capture the local motion information at those frames. In addition, we embed learnable parameters for each keyframe to represent latent motion characteristics that cannot be fully captured by velocity alone. These velocity vectors and learnable parameters are then projected together with the corresponding time values, enabling the construction of object motion features. This formulation allows us to obtain continuous motion features for objects across all frames, rather than being limited to discrete keyframes.

**Interaction Module.** The proposed Interaction module (HOI module) takes time-varying features of humans and objects as inputs and explicitly models their interactions. Let the human and object features be denoted as $F_{\text{Human}}$ and $F_{\text{Object}}$. To capture interdependencies between the two, we apply *mutual attention*, where queries, keys, and values are defined as:

$$Q_h, K_h, V_h = F_{\text{Human}}W_h^Q, \ F_{\text{Human}}W_h^K, \ F_{\text{Human}}W_h^V,$$
$$Q_o, K_o, V_o = F_{\text{Object}}W_o^Q, \ F_{\text{Object}}W_o^K, \ F_{\text{Object}}W_o^V. \tag{6-7}$$

Cross-attention is then performed in both directions, from human to object and from object to human, while incorporating a distance mask $B$ into the attention computation:

$$F'_{\text{Human}} = \text{softmax}\left(\frac{Q_h K_o^\top}{\sqrt{d}} + B\right) V_h, \quad F'_{\text{Object}} = \text{softmax}\left(\frac{Q_o K_h^\top}{\sqrt{d}} + B^\top\right) V_o. \tag{8}$$

The distance mask $B$ filters out distant objects based on their 3D spatial proximity to the human. Specifically, for each object, we compute the Euclidean distance between the object's Gaussian center $c_{\text{obj}}^{\text{world}}$ and the human pelvis position $p_{\text{pelvis}}^{\text{world}}$ in world coordinates. $B_{ij}$ encodes the relative distance between the $i$-th human token and the $j$-th object token:

$$B_{ij} = \begin{cases} -\infty & \text{if } \|c_{\text{obj}}^{\text{world}} - p_{\text{pelvis}}^{\text{world}}\| \geq \tau \\ 0 & \text{otherwise} \end{cases}, \tag{9}$$

where $\tau$ is set to the human arm length (derived from the SMPL-X model). When $B_{ij} = -\infty$, the corresponding object tokens are masked out during attention, effectively excluding non-interacting background objects.

This process yields updated features $F'_{\text{Human}}$ and $F'_{\text{Object}}$ that embed interaction cues. Finally, $F'_{\text{Human}}$ is used to regress $\Delta$SMPL-X refinements (body pose, hand pose), while $F'_{\text{Object}}$ is used to predict $\Delta G_{\text{object}}$, i.e., corrections for Gaussian-based object motion. In this way, the HOI module augments the baseline deformations (hexplane+LBS for humans and CHS for objects) with interaction-aware adjustments, enabling accurate reconstruction of human–object interaction scenes.

### 3.4 OPTIMIZATION

For background modeling, we employ the standard 3D Gaussian Splatting (3DGS) technique. During training, we isolate the background by masking out the object and human regions, allowing the static

Gaussian background to be optimized using a photometric loss. For human modeling, we regress the SMPL-X parameters by Moon et al. (2022), and incorporate an SMPL-X-based avatar model to ensure natural interaction with the object. For each frame, we extract the SMPL-X parameters and define a canonical T-pose human avatar. This canonical avatar is then deformed to match each frame using LBS. During training, image-based loss metrics such as SSIM, LPIPS, and L1-norm were utilized to compare the Gaussian renderer's output with the human region in the image.

**Object Motion Optimization**. We model the motion of objects using CHS to ensure continuity in position interpolation. A CHS is a piecewise cubic polynomial that is defined by both the positions and the first derivatives (tangents) at key points in time. By specifying the starting and ending slopes for each spline segment, CHS guarantees smooth transitions between key frames, maintaining continuity not only in the object's position but also in its velocity. In other words, the object's trajectory over time remains continuous and smooth, without abrupt jumps or changes in speed. This property is crucial for accurately modeling temporal motion in a realistic and stable manner.

**Integrated Optimization**. We train our model using an integrated optimization objective that combines multiple loss terms. Specifically, the overall loss function is formulated as:

$$\mathcal{L} = \gamma \, \mathcal{L}_{\text{object motion}} + \beta \, \mathcal{L}_{\text{human}} + \sigma \, \mathcal{L}_{\text{scene}} + \mathcal{L}_{\text{depth}}, \tag{9}$$

where $\mathcal{L}_{\text{object motion}}$, $\mathcal{L}_{\text{human}}$, and $\mathcal{L}_{\text{scene}}$ are the loss components for the object's motion, the human-related factors, and the scene context, respectively. Here, $\gamma$, $\beta$, and $\sigma$ are hyperparameters that control the relative weight of each loss term during training. By tuning these hyperparameters, we balance the influence of each component on the training objective. This integrated optimization approach ensures that the model simultaneously accounts for object motion accuracy, human interaction plausibility, and scene consistency during learning.

## 4 EXPERIMENTS

### 4.1 IMPLEMENTATION DETAILS

We use ExAvatar (Moon et al. (2024)) as the baseline human rendering model, and all hyperparameters are kept identical to those used in ExAvatar. For object deformation using splines (Ahlberg et al. (2016); De Boor & De Boor (1978)), we fix the time interval to 4 for all scenes. Training is conducted using an NVIDIA H100 GPU, taking approximately 5 hours per scene.

### 4.2 DATASETS

**HOSNeRF dataset (Liu et al. (2023)).** We use the monocular dynamic-scene dataset HOSNeRF, which captures human–object interaction scenarios. The dataset comprises recordings in six indoor and outdoor locations with six subjects interacting with objects within a single scenario. Each sequence contains 300–400 frames. For evaluation, we uniformly select 16 frames per sequence for testing and use the remaining frames for training, following HOSNeRF.

**BEHAVE dataset (Bhatnagar et al. (2022)).** We use the BEHAVE multi-view RGB-D human–object interaction dataset, but adapt it to a monocular setting by selecting a single fixed camera from the four static viewpoints for each sequence. Specifically, we curate 9 sequences covering four distinct indoor environments, five subjects, and four objects. From each sequence's raw video, we uniformly sample 300 frames. For evaluation, we uniformly select 16 frames per sequence for testing and use the remaining frames for training

**ARCTIC dataset (Fan et al. (2023)).** We use the ARCTIC hand–object interaction dataset and extend comparisons to hand–object baselines. Since HOIGS is human-centric rather than hand-only, we evaluate only sequences where the full body is visible. Specifically, we use sequences of one subject interacting with four objects. Each monocular sequence (600 frames) is split by uniformly sampling 16 frames for testing and using the rest for training.

### 4.3 QUALITATIVE RESULTS

We compare our view-synthesis results with existing Gaussian-based models, which generally outperform NeRF-based methods in rendering quality. The experimental results are visualized

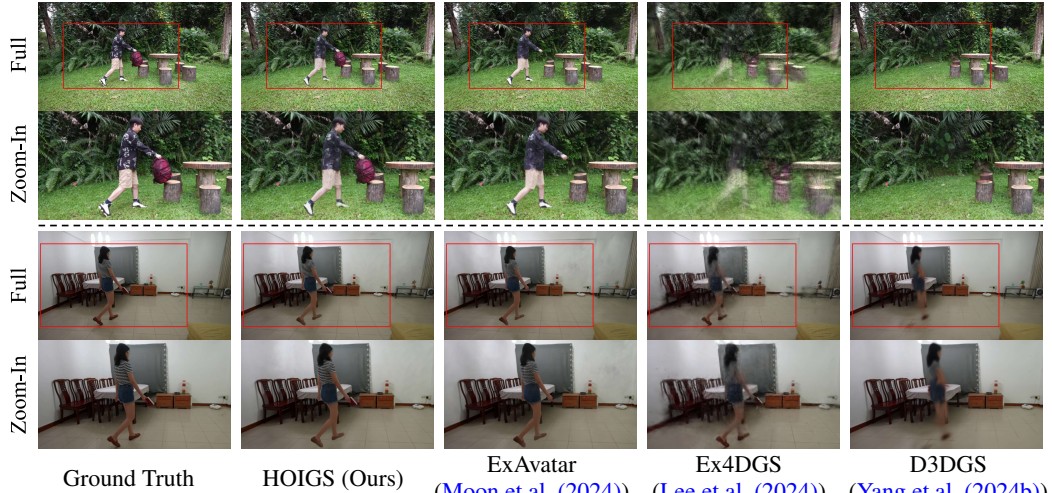

Figure 3: Qualitative comparison of reconstructed rendered view results on the HOSNeRF dataset. We display the full-frame (top) rendering and a zoom-in (bottom) of the red Region of Interest (ROI).

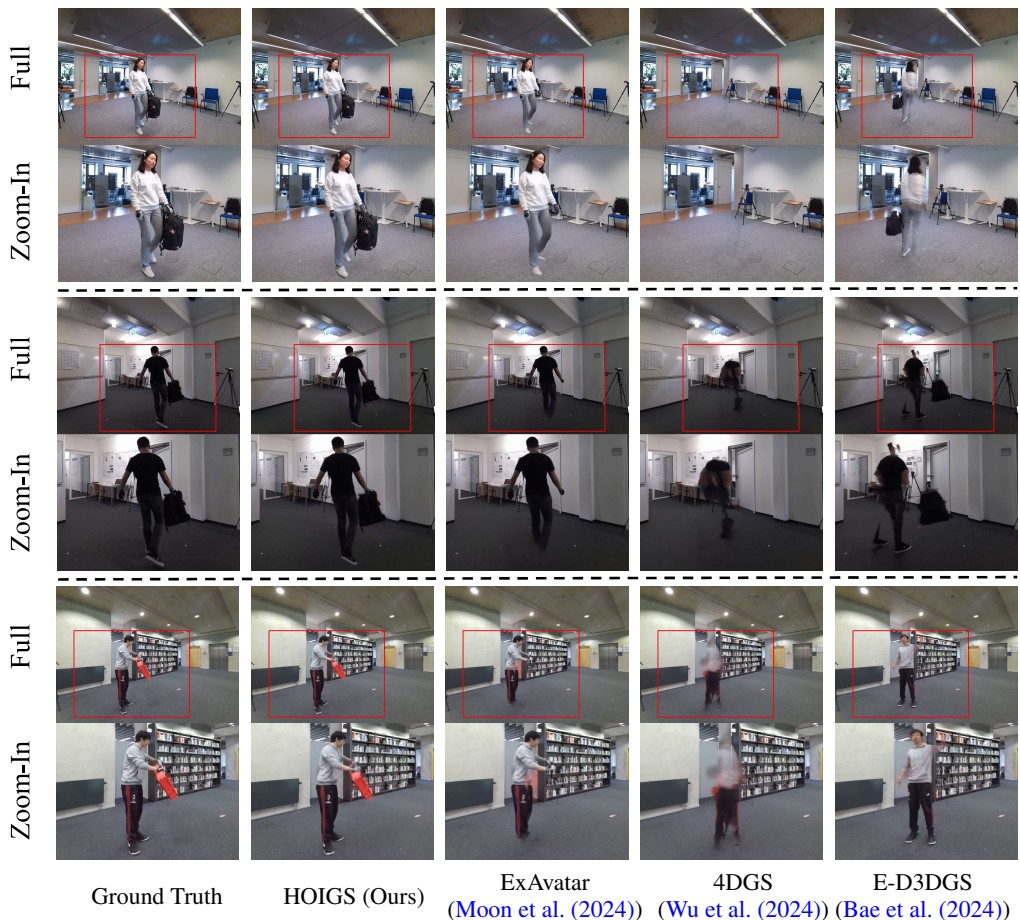

Figure 4: Qualitative comparison of reconstructed rendered view results on the BEHAVE dataset. We display the full-frame (top) rendering and a zoom-in (bottom) of the red Region of Interest (ROI).

in Fig. 3. The dynamic-scene models D3DGS (Yang et al. (2024b)) and Ex4DGS (Lee et al. (2024)) yield ghosting artifacts for both human and dynamic objects because they fail to disentangle human and object motions within complex interactions. ExAvatar (Moon et al. (2024)) reconstructs humans

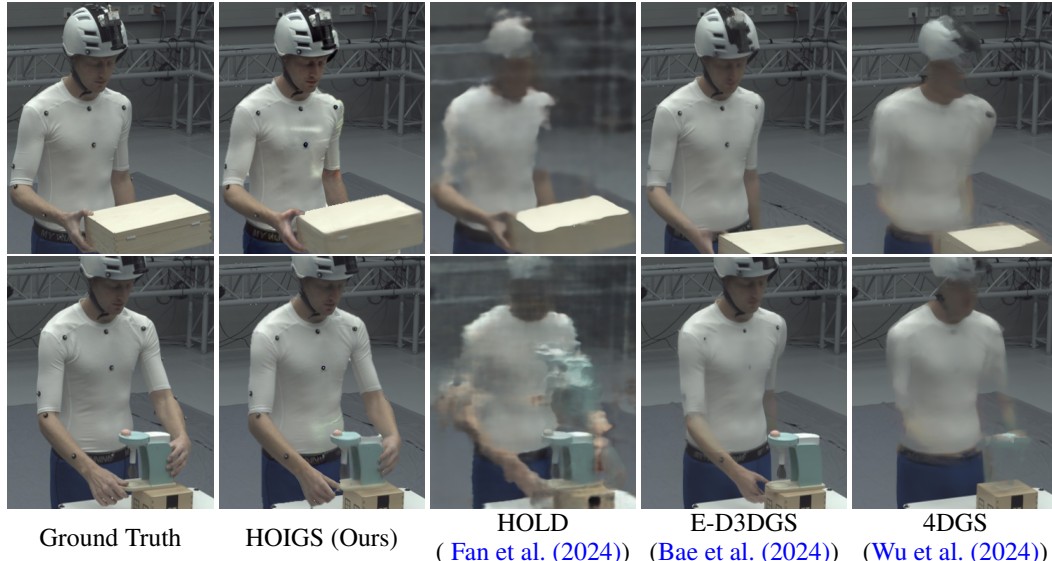

Ground Truth    HOIGS (Ours)    HOLD    E-D3DGS    4DGS
( Fan et al. (2024))    (Bae et al. (2024))    (Wu et al. (2024))

Figure 5: **Qualitative comparison of reconstructed rendered view results on the ARCTIC dataset.**

| Methods | Backpack | | Tennis | | Suitcase | | Playground | | Dance | | Lounge | |
|---|---|---|---|---|---|---|---|---|---|---|---|---|
| | PSNR↑ | LPIPS↓ | PSNR↑ | LPIPS↓ | PSNR↑ | LPIPS↓ | PSNR↑ | LPIPS↓ | PSNR↑ | LPIPS↓ | PSNR↑ | LPIPS↓ |
| K-Planes Fridovich-Keil et al. (2023) | 19.05 | 0.557 | 19.31 | 0.536 | 18.64 | 0.602 | 17.92 | 0.635 | 18.17 | 0.623 | 24.21 | 0.453 |
| D$^2$NeRF Wu et al. (2022) | 20.52 | 0.608 | 23.97 | 0.540 | 20.99 | 0.645 | 21.23 | 0.616 | 19.92 | 0.647 | 27.13 | 0.509 |
| Nerfies Park et al. (2021a) | 19.56 | 0.559 | 22.12 | 0.443 | 19.01 | 0.555 | 21.14 | 0.533 | 19.37 | 0.524 | 25.90 | 0.342 |
| HyperNeRF Park et al. (2021b) | 19.62 | 0.587 | 21.26 | 0.510 | 19.41 | 0.607 | 21.67 | 0.578 | 19.30 | 0.601 | 27.25 | 0.332 |
| NeuMan Jiang et al. (2022) | 21.21 | 0.478 | 23.17 | 0.442 | 20.84 | 0.551 | 21.46 | 0.551 | 21.19 | 0.490 | 28.40 | 0.341 |
| 4DGS Wu et al. (2024) | 24.49 | 0.192 | 26.57 | 0.162 | 17.98 | 0.460 | 24.34 | 0.222 | 21.34 | 0.212 | 30.50 | 0.067 |
| D3DGS Yang et al. (2024b) | 24.06 | 0.099 | 25.09 | 0.125 | 17.85 | 0.453 | 23.93 | 0.141 | 21.07 | 0.117 | 26.90 | 0.072 |
| E-D3DGS Bae et al. (2024) | 24.78 | 0.146 | 26.53 | 0.161 | 18.05 | 0.461 | 24.37 | 0.206 | 23.87 | 0.159 | 30.04 | 0.086 |
| Ex4DGS Lee et al. (2024) | 18.07 | 0.433 | 17.90 | 0.399 | 15.25 | 0.557 | 16.36 | 0.535 | 17.08 | 0.529 | 23.15 | 0.310 |
| ExAvatar Moon et al. (2024) | 24.15 | 0.107 | 23.57 | 0.160 | 20.32 | 0.260 | 25.30 | 0.129 | 23.32 | 0.170 | 29.43 | 0.048 |
| HOSNeRF Liu et al. (2023) | 22.56 | 0.243 | 24.15 | 0.320 | 21.74 | 0.382 | 22.67 | 0.336 | 22.63 | 0.248 | 27.74 | 0.227 |
| **HOIGS (Ours)** | 25.78 | 0.082 | 27.12 | 0.108 | 22.09 | 0.246 | 25.23 | 0.103 | 24.17 | 0.098 | 30.97 | 0.048 |

Table 1: Per-scene quantitative evaluation on the HOSNeRF dataset against baselines of our method. We color each cell as **best** and **second best** .

but does not handle dynamic objects. Our method accurately reconstructs humans and objects with temporally coherent motion, using CHS object trajectories with velocity vectors and the human backbone based on hexplane and LBS, while the HOI module further ensures contact consistency. On the ARCTIC dataset, as shown in Fig. 5, HOLD (Fan et al. (2024)) shows limited performance in full-body–object interactions, whereas HOIGS successfully reconstructs them. This is because HOLD reconstructs only hands, while HOIGS reconstructs the entire human body including the hands. On the BEHAVE dataset, as shown in Fig. 4, whereas ExAvatar suffers body–background overlap due to human misalignment in world space, our depth-based alignment ensures accurate human placement. Through qualitative results, we further confirm that our method effectively reconstructs complex human–object interactions with visually consistent outcomes.

### 4.4 QUANTITATIVE RESULTS

As shown in Tab. 1, HOIGS achieves the highest PSNR and the lowest LPIPS on the Backpack, Tennis, Suitcase, Dance, and Lounge scenarios of the HOSNeRF dataset, surpassing prior 3D Gaussian-based models in visual quality. Tab. 2 shows that on the BEHAVE dataset, it likewise attains the highest PSNR and lowest LPIPS, demonstrating effective reconstruction of complex human–object interactions from single-view input. Tab. 3 shows that on the ARCTIC dataset, our method outperforms the hand–object model HOLD (Fan et al. (2024)). Unlike HOLD, our model reconstructs complex full-body geometry while simultaneously capturing interactions with dynamic objects.

| Methods | Backpack_1 | | Plasticcontainer_1 | | Plasticcontainer_2 | | Suitcase_1 | |
|---|---|---|---|---|---|---|---|---|
| | PSNR↑ | LPIPS↓ | PSNR↑ | LPIPS↓ | PSNR↑ | LPIPS↓ | PSNR↑ | LPIPS↓ |
| 4DGS Wu et al. (2024) | 21.81 | 0.076 | 22.92 | 0.072 | 26.37 | 0.081 | 26.66 | 0.071 |
| E-D3DGS Bae et al. (2024) | 19.99 | 0.086 | 20.15 | 0.086 | 24.75 | 0.078 | 25.85 | 0.058 |
| ExAvatar Moon et al. (2024) | 27.86 | 0.041 | 29.96 | 0.042 | 30.11 | 0.038 | 30.86 | 0.032 |
| **HOIGS (Ours)** | 31.79 | 0.031 | 33.10 | 0.032 | 32.39 | 0.034 | 34.58 | 0.028 |
| Methods | Backpack_2 | | Plasticcontainer_3 | | Backpack_3 | | Trashbin | |
| | PSNR↑ | LPIPS↓ | PSNR↑ | LPIPS↓ | PSNR↑ | LPIPS↓ | PSNR↑ | LPIPS↓ |
| 4DGS Wu et al. (2024) | 24.59 | 0.085 | 24.60 | 0.087 | 23.43 | 0.090 | 26.07 | 0.082 |
| E-D3DGS Bae et al. (2024) | 23.72 | 0.074 | 23.81 | 0.070 | 22.07 | 0.079 | 25.56 | 0.062 |
| ExAvatar Moon et al. (2024) | 26.47 | 0.054 | 26.71 | 0.056 | 25.78 | 0.038 | 29.81 | 0.029 |
| **HOIGS (Ours)** | 30.17 | 0.044 | 29.38 | 0.046 | 29.05 | 0.030 | 31.62 | 0.023 |

Table 2: Per-scene quantitative evaluation on the **BEHAVE** dataset against baselines. We color each cell as **best** and **second best**.

| Methods | Capsulemachine | | Box | | Espressomachine | | Mixer | |
|---|---|---|---|---|---|---|---|---|
| | PSNR↑ | LPIPS↓ | PSNR↑ | LPIPS↓ | PSNR↑ | LPIPS↓ | PSNR↑ | LPIPS↓ |
| 4DGS Wu et al. (2024) | 26.15 | 0.124 | 22.22 | 0.182 | 21.80 | 0.196 | 23.21 | 0.166 |
| E-D3DGS Bae et al. (2024) | 25.10 | 0.089 | 20.60 | 0.153 | 19.50 | 0.227 | 22.14 | 0.139 |
| HOLD Fan et al. (2024) | 25.52 | 0.522 | 24.72 | 0.494 | 23.52 | 0.547 | 23.35 | 0.540 |
| **HOIGS (Ours)** | 27.05 | 0.069 | 23.50 | 0.124 | 25.29 | 0.079 | 24.59 | 0.095 |

Table 3: Per-scene quantitative evaluation on the ARCTIC dataset against baselines of our method. We color each cell as **best** and **second best**.

| | Avg (6 scenes) | |
|---|---|---|
| | PSNR↑ | LPIPS↓ |
| w/o CHS deformation (using MLP) | 24.52 | 0.154 |
| Baseline deformation | 25.01 | 0.130 |
| w/o human feature | 25.67 | 0.119 |
| w/o HOI module | 25.24 | 0.128 |
| HOIGS (**Ours**) | **25.89** | **0.114** |

Table 4: Ablation studies on the HOSNeRF dataset using our method. The **best** results are highlighted.

## 4.5 ABLATION STUDY

We conduct ablation studies to validate the effectiveness of the proposed method. As shown in Tab. 4, modeling object deformation with a simple MLP yields the lowest performance, while our CHS-based baseline deformation improves PSNR by 0.5, demonstrating its superiority. Removing the HOI module and applying only velocity further results in a 0.6 drop in PSNR compared to the full model, confirming the necessity of explicitly modeling human–object interactions. Finally, replacing the time-varying hexplane features with simple parameter embeddings for the human features leads to a 0.2 decrease in PSNR, highlighting the effectiveness of our human feature design.

## 5 CONCLUSION

We presented HOIGS, a novel framework for reconstructing dynamic scenes with explicit modeling of human–object interactions from monocular videos. By combining hexplane-based human deformation, spline-based object motion, and an interaction-aware HOI module, our method achieves stable and accurate reconstruction even in challenging scenarios with contact and manipulation. In particular, the explicit treatment of human-object interactions enables our framework not only to recover realistic human geometry but also to faithfully capture object dynamics and their mutual influences, which have been largely overlooked in prior works. Extensive experiments on HOSNeRF, BEHAVE, and ARCTIC datasets demonstrate that HOIGS outperforms state-of-the-art human-centric and 4D Gaussian approaches in both visual quality and consistency, highlighting its effectiveness in advancing realistic modeling of complex human–object interactions.

**Limitations and future works.** While our framework handles typical dynamic motions well, it struggles under minimal camera movement, where COLMAP-based pose and point cloud estimation becomes unreliable. This often leads to rendering artifacts. Future work may improve robustness in such low-baseline settings by jointly optimizing camera poses during training.

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

# 6 APPENDIX

## STATEMENT ON THE USE OF LARGE LANGUAGE MODELS

In the interest of transparency and in compliance with the ICLR 2026 guidelines, we report that a large language model (LLM) was used to assist in the refinement of this paper's text.

**Scope of Use.** The model's role was strictly limited to that of a writing assistant. Its contributions include:

- Correcting grammatical errors, spelling, and punctuation.
- Improving sentence structure and flow for enhanced clarity.
- Refining word choices for greater precision and conciseness.

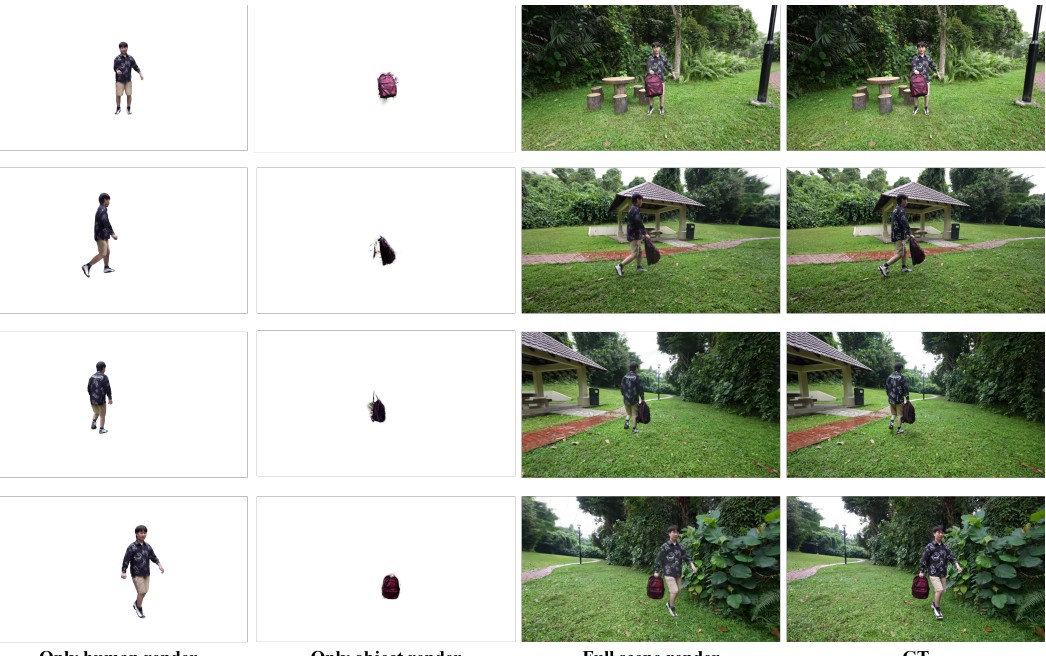

| Only human render | Only object render | Full scene render | GT |

Figure 6: **Decomposed scene reconstruction.** Visualization of individual scene components demonstrating geometric integrity in occluded regions. From left to right: human-only rendering, object-only rendering, full scene rendering, and ground truth. Each row shows a different frame from the sequence, highlighting that both human and object maintain coherent geometry even during close contact and occlusion.

## 6.1 DECOMPOSED VISUALIZATION

**Component-level Reconstruction Quality.** To address concerns about reconstruction quality in occluded regions, we provide decomposed visualizations that isolate individual scene components. As shown in Figure 6, we render the human and object separately to demonstrate that each component maintains geometric integrity even in regions with heavy occlusion or contact.

**Human-only Rendering (Column 1):** The isolated human reconstruction shows coherent body geometry throughout the sequence, including regions that were occluded by the object (backpack) in the original footage. This demonstrates that our hexplane-based human deformation successfully captures the complete body structure without artifacts from the interacting object.

**Object-only Rendering (Column 2):** The object is reconstructed as a distinct, stable entity with well-defined geometry. Unlike single-field approaches that often produce fused or melted geometry

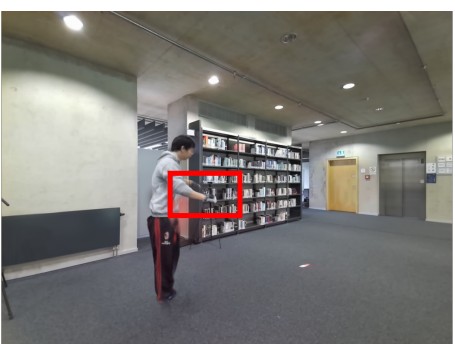 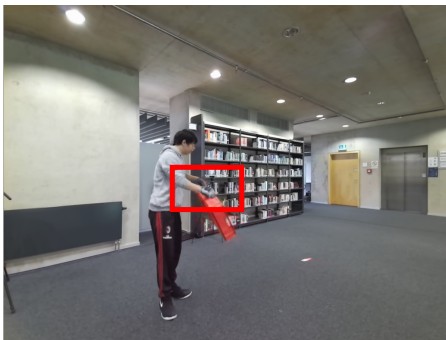

**ExAvatar [ECCV 24]**          **Ours**

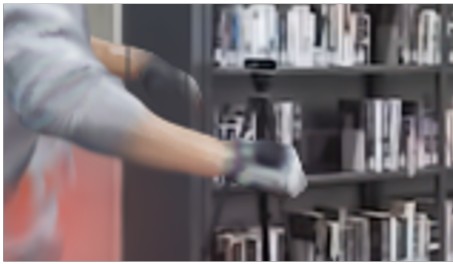 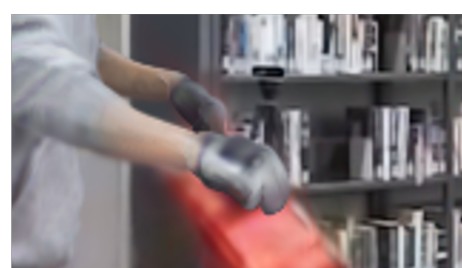

**ExAvatar [ECCV 24]**          **Ours**

Figure 7: **Qualitative comparison of human pose refinement on the BEHAVE dataset.** Visual comparison between ExAvatar and our method (HOIGS). The red boxes highlight the interaction regions (hands and objects). Our HOI module explicitly refines the hand and forearm poses by leveraging object motion features, leading to accurate contact modeling, whereas ExAvatar exhibits misalignment in these interaction-rich regions.

at contact points, our CHS-based object deformation maintains clear boundaries and structural consistency throughout the interaction.

**Full Scene Rendering (Column 3):** The combined rendering seamlessly integrates both components and closely matches the ground truth, confirming that our explicit modeling of human-object interactions through the HOI module enables accurate disentanglement while preserving realistic appearance.

These results validate that HOIGS does not simply overfit the combined RGB appearance but genuinely learns separate geometric representations for humans and objects. The clean separation at contact boundaries and the preservation of geometry in occluded regions demonstrate the effectiveness of our approach in handling complex interaction scenarios.

### 6.2 QUANTITATIVE EVALUATION OF HUMAN POSE ACCURACY

**Geometric Fidelity Analysis**. We acknowledge that rendering metrics alone are insufficient to fully validate the geometric fidelity of complex human–object interactions. To address this, we conducted additional evaluations on human pose accuracy using the BEHAVE dataset. We compare our method against ExAvatar using PA-MPJPE (Procrustes Aligned Mean Per Joint Position Error) and PA-PVE (Procrustes-Aligned Per Vertex Error).

**Effect of the HOI Module on Pose Refinement**. Our method explicitly models the mutual dependency between the human and the object. The HOI module leverages a cross-attention mechanism to use object motion features as contextual cues to refine human features. Specifically, as described in Eq. 14 of the main paper, the module predicts refinement offsets $\Delta$SMPL-X for the body and hands. This capability allows the network to correct the human pose—even under partial occlusion—by inferring the likely body configuration from the object's trajectory.

| Model | Backpack1 | Plasticcontainer1 | Plasticcontainer2 | Suitcase1 |
|---|---|---|---|---|
| ExAvatar | 0.4196 / 0.4628 / 0.3687 | 0.3875 / 0.4282 / 0.3563 | 0.3094 / 0.3145 / 0.2897 | 0.2654 / 0.2957 / 0.2505 |
| HOIGS (Ours) | 0.4177 / 0.4539 / 0.3656 | 0.2964 / 0.3344 / 0.2863 | 0.2973 / 0.3120 / 0.2837 | 0.2438 / 0.2649 / 0.2352 |
| **Model** | **Backpack2** | **Plasticcontainer3** | **Backpack3** | **Trashbin** |
| ExAvatar | 0.2690 / 0.3135 / 0.2488 | 0.3293 / 0.3298 / 0.2970 | 0.2177 / 0.2494 / 0.2092 | 0.2294 / 0.2597 / 0.2156 |
| HOIGS (Ours) | 0.2629 / 0.3068 / 0.2438 | 0.3270 / 0.3265 / 0.2948 | 0.2110 / 0.2380 / 0.2020 | 0.2263 / 0.2550 / 0.2126 |

Table 5: Unified quantitative evaluation on the BEHAVE dataset. The values in each cell correspond to **PA-MPJPE / PA-MPJPE (Hand/Forearm) / PA-PVE**. HOIGS consistently outperforms the baseline across these metrics, particularly in interaction-critical regions.

| Method Combinations | Backpack | | Tennis | | Suitcase | | Playground | | Dance | | Lounge | | Average | |
|---|---|---|---|---|---|---|---|---|---|---|---|---|---|---|
| | PSNR↑ | LPIPS↓ | PSNR↑ | LPIPS↓ | PSNR↑ | LPIPS↓ | PSNR↑ | LPIPS↓ | PSNR↑ | LPIPS↓ | PSNR↑ | LPIPS↓ | PSNR↑ | LPIPS↓ |
| Samurai + MetricV2 | 25.78 | 0.082 | 27.12 | 0.108 | 22.09 | 0.246 | 25.23 | 0.103 | 24.17 | 0.098 | 30.97 | 0.048 | 25.89 | 0.114 |
| Samurai + Video Depth Anything | 25.85 | 0.080 | 27.18 | 0.106 | 22.15 | 0.241 | 25.28 | 0.102 | 24.22 | 0.096 | 31.05 | 0.046 | 25.96 | 0.112 |
| Samurai + DepthCrafter | 25.72 | 0.088 | 27.08 | 0.108 | 22.06 | 0.249 | 25.20 | 0.109 | 24.08 | 0.099 | 30.93 | 0.048 | 25.85 | 0.117 |
| TrackAnything + MetricV2 | 25.72 | 0.086 | 27.05 | 0.109 | 22.03 | 0.246 | 25.18 | 0.106 | 24.15 | 0.100 | 30.93 | 0.052 | 25.84 | 0.116 |
| SAMv2 + Video Depth Anything | 26.01 | 0.076 | 27.38 | 0.103 | 22.33 | 0.241 | 25.47 | 0.100 | 24.42 | 0.095 | 31.20 | 0.041 | 26.14 | 0.109 |
| MaskRCNN + MetricV2 | 25.33 | 0.099 | 26.67 | 0.125 | 21.66 | 0.257 | 24.78 | 0.117 | 23.69 | 0.110 | 30.52 | 0.065 | 25.44 | 0.129 |

Table 6: Sensitivity analysis of HOIGS on the HOSNeRF dataset using different combinations of segmentation and depth estimation priors. The results demonstrate the robustness of our method, with consistent performance across various modern priors and strong performance even with older baselines (MaskRCNN).

**Quantitative Results**. Table 5 summarizes the evaluation on the BEHAVE dataset. We report the average PA-MPJPE and PA-PVE across all test frames. Additionally, we provide a specific analysis for Hand and Forearm joints, which are the most critical regions for interaction tasks. As shown in Table 5, HOIGS consistently outperforms the baseline. Notably, we observe a larger performance gain in the **PA-MPJPE (Hand/Forearm joints)**. This indicates that our HOI module effectively refines the poses of interaction-related body parts, resulting in physically more accurate reconstructions compared to ExAvatar, which lacks mutual feedback between the human and the object. Please refer to the per-sequence detailed tables at the bottom of the appendix.

## 6.3 SENSITIVITY ANALYSIS ON EXTERNAL MODULES

**Robustness to External Priors.** To address concerns regarding the reliance on external modules, we conducted a sensitivity analysis on the HOSNeRF dataset by evaluating our framework with various combinations of segmentation (e.g., Samurai Yang et al. (2024a), SAMv2 Ravi et al. (2024), MaskRCNN Massa & Girshick (2018), TrackAnything Yang et al. (2023a)) and depth estimation (e.g., Video Depth Anything Chen et al. (2025), MetricV2 Hu et al. (2024b), DepthCrafter Hu et al. (2025)) models. As shown in Table 6, HOIGS maintains highly consistent performance (Avg PSNR 25.8–26.1) across different modern priors, demonstrating that our method is robust to variations in preprocessing quality. Notably, even when employing the standard, older baseline of MaskRCNN combined with MetricV2, our model achieves an average PSNR of 25.44. This performance remains significantly higher than the state-of-the-art human-centric baseline, ExAvatar (Avg PSNR 24.35), and the 4DGS baseline, Ex4DGS (Avg PSNR 17.97).

## 6.4 COMPUTATIONAL COMPLEXITY AND RUNTIME ANALYSIS

**Runtime Performance**. We evaluate the computational efficiency of our method on the HOSNeRF dataset using a single NVIDIA H100 GPU. As shown in Table 7, our method achieves an inference speed of **44.27 FPS**. While this is slightly lower than 4DGS Wu et al. (2024) (61.04 FPS), it remains comparable to Ex4DGS Lee et al. (2024) (46.38 FPS) and outperforms D3DGS Yang et al. (2024b) (37.79 FPS). This result confirms that the inclusion of the HOI attention mechanism does not create a significant bottleneck, allowing our method to comfortably support real-time applications.

**Complexity Analysis**. The efficiency of our HOI module stems from the token-based architectural design. The cross-attention is computed between $M$ human part tokens (where $M = 16$ is fixed) and $N$ object Gaussian tokens. Unlike standard self-attention which scales quadratically ($O(N^2)$), our cross-attention scales linearly ($O(M \cdot N)$) with respect to the number of object Gaussians.

| Methods | Training Time | Inference Speed (FPS) |
|---|---|---|
| 4DGS Wu et al. (2024) | 40 min | 61.04 |
| Ex4DGS Lee et al. (2024) | 2 hr 30 min | 46.38 |
| D3DGS Yang et al. (2024b) | 3 hr | 37.79 |
| E-D3DGS Bae et al. (2024) | 2 hr | 54.71 |
| **HOIGS (Ours)** | 5 hr | 44.27 |

Table 7: Runtime performance comparison on the HOSNeRF dataset. We report the approximate training time per scene and the inference speed in Frames Per Second (FPS). Our method maintains real-time performance (>30 FPS) despite the added complexity of interaction modeling.

| Methods | Backpack | | Tennis | | Suitcase | | Playground | | Dance | | Lounge | | Average | |
|---|---|---|---|---|---|---|---|---|---|---|---|---|---|---|
| | PSNR↑ | LPIPS↓ | PSNR↑ | LPIPS↓ | PSNR↑ | LPIPS↓ | PSNR↑ | LPIPS↓ | PSNR↑ | LPIPS↓ | PSNR↑ | LPIPS↓ | PSNR↑ | LPIPS↓ |
| MASt3R Prior | 23.51 | 0.135 | 25.25 | 0.121 | 22.40 | 0.197 | 24.65 | 0.074 | 23.63 | 0.115 | 28.99 | 0.057 | 24.59 | 0.128 |
| Depth Recon Prior | 21.63 | 0.142 | 25.65 | 0.122 | 22.13 | 0.230 | 25.24 | 0.103 | 24.08 | 0.123 | 28.95 | 0.095 | 25.36 | 0.136 |
| **Diffusion Prior (Ours)** | 23.70 | 0.082 | 27.13 | 0.112 | 22.96 | 0.235 | 25.63 | 0.123 | 24.17 | 0.093 | 29.97 | 0.043 | 25.89 | 0.114 |

Table 8: Quantitative ablation study on Object Priors using the HOSNeRF dataset. We evaluate the effectiveness of our Diffusion Prior against MASt3R and Depth Reconstruction priors.

Furthermore, we utilize compact 32-dimensional embeddings for object motion features, which minimizes the memory footprint and matrix multiplication overhead during the forward pass.

**Training Cost Justification**. We acknowledge that our training time (~5 hours) is longer than the baselines. This is a deliberate trade-off to prioritize physical plausibility and interaction accuracy. Explicitly modeling mutual dependencies and backpropagating gradients through the attention mechanism requires more iterations. However, this cost is strictly confined to the offline training phase, ensuring that the final online user experience remains real-time.

## 6.5 ADDITIONAL ABLATION STUDIES

**Impact of Object Diffusion Prior.** To validate the effectiveness of our design choice, we investigate the impact of different geometric priors on the final reconstruction quality. We compare our proposed method, which utilizes a generative Diffusion Prior, against two alternative initialization strategies:
(1) MASt3R Prior: Initialization using MASt3RDuisterhof et al. (2025), a state-of-the-art dense matching and reconstruction model.
(2) Depth Reconstruction Prior: Initialization using standard monocular metric depth estimation.

Table 8 presents the quantitative comparison on the HOSNeRF dataset. Our method equipped with the Diffusion Prior achieves the highest average reconstruction quality (25.89 PSNR), outperforming the MASt3R prior (24.59 PSNR) and the Depth prior (25.36 PSNR). While discriminative approaches like MASt3R or metric depth estimation rely heavily on visible cues, they often struggle to reconstruct accurate geometry in the presence of heavy occlusions, a common occurrence in human-object interaction scenarios (e.g., hands covering objects). In contrast, the Diffusion Prior leverages generative knowledge to plausibly complete 3D geometry even in occluded or unseen regions. This holistic geometric initialization provides a more robust starting point for our Cubic Hermite Spline (CHS) deformation, leading to sharper rendering and more stable tracking throughout the dynamic sequence.

## 6.6 FEATURE EXTRACTION

**Object feature**. As shown in Fig. 8(a), we extract object features by leveraging the velocity vectors and embedding parameters of Gaussians at key frames. As shown in Fig. 8(b), each key frame's velocity vector is applied to the CHS and jointly optimized with the baseline deformation as input features for the HOI module. In addition, a 29-dimensional learnable parameter is embedded for each key frame Gaussian, which is concatenated with the velocity vector to form the feature representation. The interpolated Gaussian features produced by CHS are then combined with the concatenated feature and time information, and projected through a shallow MLP, resulting in a 32-dimensional feature vector.

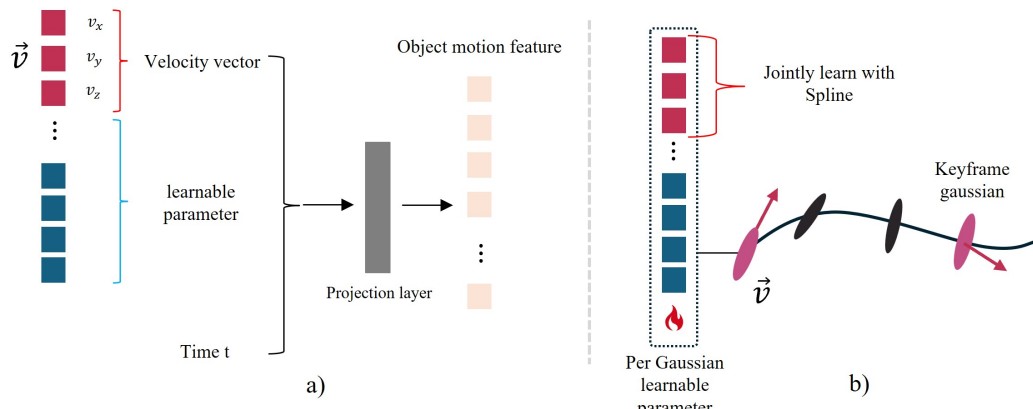

Figure 8: **Object feature extraction.** Extraction of object motion features using the embedded parameters and velocity vectors of each key frame.

**Human feature**. Fig. 9 illustrates the process of human feature extraction. We divide the SMPL-X model into 16 body parts and learn features corresponding to each part. Temporal features are sampled from the hexplane at SMPL-X vertices, where each feature at time $t$ is obtained based on the coordinates $(x_t, y_t, z_t)$. For each body part, the features of its associated vertices are averaged to form the part-specific representation $F_{\text{human}}$:

$$F_{\text{part}} = \frac{1}{N} \sum_{i \in \text{part}} f_i(x_t, y_t, z_t), \tag{10}$$

where $N$ denotes the number of vertices belonging to the part. As a result, 16 part features, including head, torso, arms, and legs, are obtained and used as inputs to the HOI module. This design captures temporally varying dynamic representations while preserving semantically meaningful features for individual body parts.

### 6.7 HOI MODULE NETWORK DETAIL

As shown in Fig. 10, the proposed HOI module takes the time-varying features of humans and objects as inputs and explicitly models their interactions. Let the human feature be denoted as $F_{\text{Human}} \in \mathbb{R}^{N_h \times d}$ and the object feature as $F_{\text{Object}} \in \mathbb{R}^{N_o \times d}$, where $N_h$ and $N_o$ are the numbers of feature tokens for human and object respectively, and $d$ is the feature dimension.

To capture interdependencies between the two modalities, we apply a *mutual-attention* mechanism. Specifically, queries ($Q$), keys ($K$), and values ($V$) are obtained via learnable linear projections:

$$Q_h = F_{\text{Human}}W_h^Q, \quad K_o = F_{\text{Object}}W_o^K, \quad V_o = F_{\text{Object}}W_o^V, \tag{11}$$

$$Q_o = F_{\text{Object}}W_o^Q, \quad K_h = F_{\text{Human}}W_h^K, \quad V_h = F_{\text{Human}}W_h^V, \tag{12}$$

where $W_h^Q, W_h^K, W_h^V, W_o^Q, W_o^K, W_o^V \in \mathbb{R}^{d \times d}$ are learnable projection matrices.

Cross-attention is then computed in both directions: from human to object and from object to human. To enforce spatial priors, a distance mask $B \in \mathbb{R}^{N_h \times N_o}$ is added to the attention logits, where $B_{ij}$ encodes the relative distance between the $i$-th human token and the $j$-th object token. The resulting attention maps are defined as:

$$A_{h \leftarrow o} = \text{softmax}\left(\frac{Q_h K_o^\top}{\sqrt{d}} + B\right), \quad A_{o \leftarrow h} = \text{softmax}\left(\frac{Q_o K_h^\top}{\sqrt{d}} + B^\top\right). \tag{13}$$

Using these attention weights, the updated features are obtained as:

$$F'_{\text{Human}} = A_{h \leftarrow o}V_h, \quad F'_{\text{Object}} = A_{o \leftarrow h}V_o. \tag{14}$$

The updated human feature $F'_{\text{Human}}$ is then fed into a small MLP head to regress the refinement terms of SMPL-X parameters:

$$\Delta\text{SMPL-X} = \{\Delta\theta_{\text{body}}, \ \Delta\theta_{\text{hand}}\}, \tag{15}$$

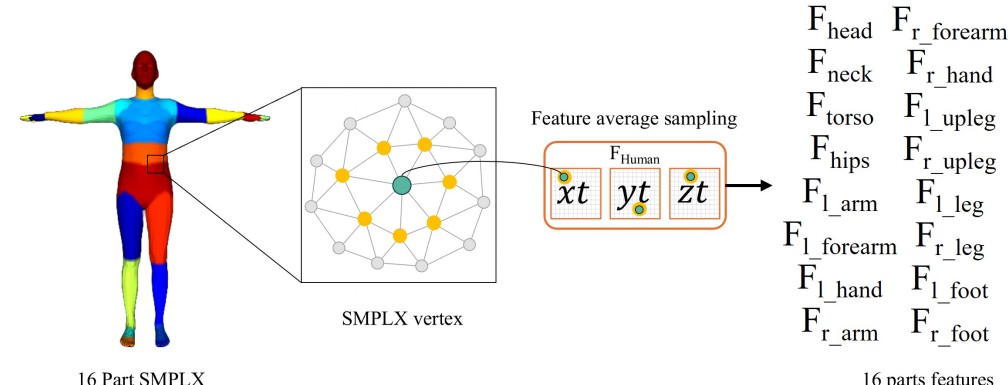

Figure 9: **Human feature extraction.**

where $\Delta\theta_{\text{body}}$ and $\Delta\theta_{\text{hand}}$ denote pose corrections for body and hands. Similarly, the updated object feature $F'_{\text{Object}}$ is used to regress Gaussian-based object motion corrections:

$$\Delta G_{\text{object}} \in \mathbb{R}^{N_o \times 3}, \tag{16}$$

which represent displacement vectors applied to object Gaussians.

In this way, the HOI module augments the baseline deformations (hexplane+LBS for humans and CHS for objects) with interaction-aware refinements, enabling accurate reconstruction of complex human–object interaction scenes.

## 6.8 OBJECTIVE FUNCTION DETAILS

The overall loss function of our model is defined as follows:

$$L = \gamma L_{\text{object motion}} + \beta L_{\text{human}} + \sigma L_{\text{scene}} + L_{\text{depth}}, \tag{17}$$

where $L_{\text{object motion}}$, $L_{\text{human}}$, and $L_{\text{scene}}$ correspond to losses for object motion, human modeling, and scene context, respectively. The weights $\gamma$, $\beta$, and $\sigma$ control the relative importance of each loss term and are specifically set to 1.0, 0.5, and 0.25, respectively. In our approach, these three terms are optimized simultaneously to consistently model the interactions between humans and objects.

**Human Loss details**
The $L_{\text{human}}$ term consists of losses related to human representation using the SMPL-X (Pavlakos et al. (2019)) model. Specifically, it includes the reprojection error between the 3D human joint positions and detected 2D keypoints in images, a mesh-based face loss enhancing the consistency of facial geometry and texture, and a Laplacian regularization term. Additionally, there is an L1 loss ($L_{\text{smplx}}$) between the optimized SMPL-X parameters and the frame-wise initial SMPL-X parameters obtained by a regressor. These loss terms are directly adopted from previous methods such as ExAvatar (Moon et al. (2024)), without modifications. For example, the face loss optimizes the consistency between rendered facial images and actual facial images, ensuring geometry-texture coherence. Laplacian regularization is applied to enhance the stability of human body shape. Further details can be found in the referenced research.

Formally, the human loss is given by:

$$L_{\text{human}} = L_{\text{kpt}} + L_{\text{face}} + L_{\text{reg}} + 0.1 \times L_{\text{smplx}}, \tag{18}$$

**Scene Loss details**
The $L_{\text{scene}}$ term is a photometric loss focusing on the background regions of the entire scene, following the image similarity-based loss used in existing 3D Gaussian Splatting (Kerbl et al. (2023)) (3DGS) methods. Specifically, a pre-trained human/object segmentation model is employed to mask out human and object regions in the images, optimizing the background Gaussians for the remaining pixels only. This involves minimizing the difference between the rendered result and the background pixels

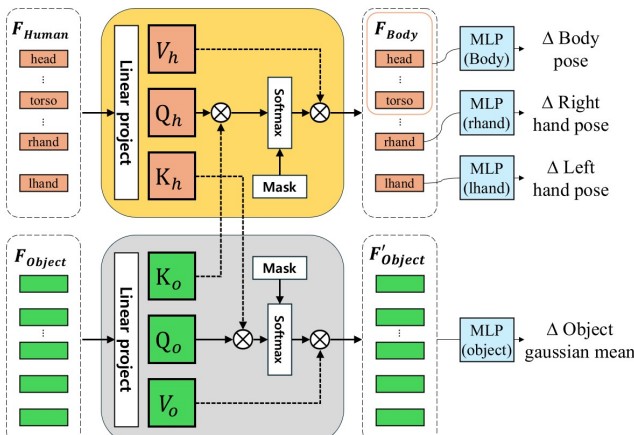

Figure 10: **Detailed HOI network.** The proposed architecture for estimating human-object interactions, leveraging features from human body parts and object Gaussian representations. The model takes as input human part features and per-Gaussian object features, processes them through bidirectional attention mechanisms to incorporate mutual contextual information, and outputs predictions for SMPL-X parameters per body part along with offset adjustments for object Gaussian centers.

excluding the segmented human and object areas. Occlusions frequently occur during interactions between human hands and objects, causing inconsistencies in masks. By optimizing humans, objects, and backgrounds simultaneously, our method effectively mitigates these boundary inconsistencies.

The scene loss is explicitly defined as:

$$L_{\text{scene}} = 0.8 \times L_1(I_{\text{gt}}, I_{\text{render}}) + 0.2 \times L_{\text{D-SSIM}}(I_{\text{gt}}, I_{\text{render}}), \tag{19}$$

**Object Loss details**

The $L_{\text{object}}$ term is a photometric loss that focuses exclusively on the object regions within the scene. We render only the segmented object areas and compute the loss solely on these regions. A pre-trained object segmentation model is employed to isolate object masks in the input images. The object loss encourages accurate reconstruction and appearance consistency for moving objects, which often undergo significant deformation and motion. By supervising only the object regions, this loss helps to refine the geometry and texture of the object-specific Gaussians without being influenced by background or human-related elements.

The object loss is defined as:

$$L_{\text{object motion}} = 0.8 \times L_1(I_{\text{gt}}, I_{\text{obj}}) + 0.2 \times L_{\text{D-SSIM}}(I_{\text{gt}}, I_{\text{obj}}). \tag{20}$$

