# OpenReview forum: "HOIGS: Human-Object Interaction Gaussian Splatting from Monocular Videos"
_ICLR.cc/2026/Conference — Submitted to ICLR 2026_

### Official Review · Reviewer_PjSB · 2025-10-25

**Soundness:** 3
**Presentation:** 2
**Contribution:** 3
**Rating:** 6
**Confidence:** 4

**Summary:**

This paper proposes a method for reconstructing human-object interaction based on the popular 3D Gaussian Splatting from monocular videos. Existing 3DGS methods fail to reconstruct interaction area as the interaction area, which is critical component of reconstructing human-object interaction. The method utilizes hexplane for human and Cubic Hermite Spline for object and propose HOI module based on cross-attention to model the human-object interaction. This leads to better performance in human-object interaction datasets like BEHAVE.

**Strengths:**

1. The paper addresses critical problem of the inability to accurately reconstruct human-object interaction by existing 1) general 3DGS methods, 2) human-oriented 3DGS methods. The qualitative results demonstrate that their approach solves this problem meaningfully.
2. The methodology section is detailed enough for researchers that are familiar with 3DGS or 3D human reconstruction to implement most of the components of the paper based solely on the paper.
3. The performance gap between existing SOTA methods and HOIGS is significant in highly interacting datasets like BEHAVE.

**Weaknesses:**

1. The qualitative results in the paper only show the aggregate 3D reconstruction of the human, object, and scene. However, it seems highly likely that the 3D human or object components may not be well reconstructed in occluded regions. Qualitative visualizations focusing solely on the 3D human and object, or even only the 3D human (as in Figure 8 of ExAvatar), would be critical to properly assess the performance of HOIGS.
2. The paper describes that the 3D object is first reconstructed using a diffusion prior based on SDS loss (similar to Zero-123), and this initial model is used as the initialization for 3D Gaussians. This pipeline appears rather complex and may not be the most straightforward approach for 3D object reconstruction. Could the authors clarify why this particular design was chosen? It would strengthen the paper to include comparisons with alternative approaches, such as SDF-based methods (e.g., Vid2Avatar) or point-based methods (e.g., Dust3r), specifically for the object reconstruction task.
3. Section 3.1 introduces a human deformation module. What would happen if this component were replaced with ExAvatar? Is there a reason the authors did not directly adopt ExAvatar for human deformation?
4. It would be interesting to see whether the contact-based masking strategy from CONTHO (CVPR 2024, Nam et al.) could improve performance compared to the current cross-attention mechanism in HOIGS. In particular, the CRFormer module in CONTHO might serve as a drop-in replacement for the HOI module.
5. The paper mentions a “distance mask B” in the HOI module. Could the authors elaborate on how this mask is generated and what its specific purpose is in the cross-attention computation?
6. A quantitative comparison with existing 3D human–object reconstruction methods such as PHOSA (ECCV 2020, Zhang et al.), CHORE (ECCV 2022, Xie et al.), and CONTHO (CVPR 2024, Nam et al.) would be beneficial to analyze the 3D reconstruction quality of results by HOIGS.
7. How are human-based 3D Gaussian reconstruction methods like ExAvatar fairly compared against HOIGS, given that they focus solely on 3D textured human reconstruction? Clarifying the evaluation scope and fairness of comparison would be helpful.
8. Section 3.4 describes the reconstruction of the background, while human and object reconstructions are modeled separately. How are these separately reconstructed components aligned or placed in a common 3D coordinate space?
9. In Figure 4, the qualitative results for 4DGS and E-D3DGS appear to correspond to different time frames, as the wooden box held by the human is in a different position. Is this discrepancy due to using different frames, or is it an inherent limitation of those methods?
10. The paper mentions the concept of “contact” multiple times, yet no contact-specific methodology is presented. Clarifying whether contact information is explicitly modeled, inferred, or simply discussed conceptually would improve the technical completeness of the paper.

**Questions:**

Listed as part of weaknesses.

**Details Of Ethics Concerns:**

Not applicable.

---

> ### Author Response · Authors · 2025-11-21
> **Official Comment by Authors**
>
> ### **Weakness 1 (Reviewer Comment)**
>
> *The qualitative results in the paper only show the aggregate 3D reconstruction of the human, object, and scene. However, it seems highly likely that the 3D human or object components may not be well reconstructed in occluded regions. Qualitative visualizations focusing solely on the 3D human and object, or even only the 3D human (as in Figure 8 of ExAvatar), would be critical to properly assess the performance of HOIGS.*
>
> ---
>
> ### **Response**
>
> We thank the reviewer for highlighting the importance of evaluating disentangled reconstruction quality. We agree that validating the geometry of individual components (human vs. object) is critical to ensure the model is not simply overfitting the combined RGB appearance, especially in regions with heavy occlusion or contact.
>
> ---
>
> ### **1. Decomposed Visualization (Figure in supplementary material)**
>
> As requested, we have generated decomposed renderings of the scene, isolating the human and object components. Please refer to the qualitative results shown in the image named “reviewer_PjSB (reviewer4) _ weakness1” on the supplementary material, which correspond to the results provided here.
>
> - **Column 1 – Only Human Render:**
>   Demonstrates that the human geometry remains coherent and intact, even in regions where the object (bag) was blocking the body.
>
> - **Column 2 – Only Object Render:**
>   Shows that the object is reconstructed as a distinct entity with stable geometry, rather than being fused with the human body.
>
> - **Column 3 – Full Scene vs. GT:**
>   Confirms that the seamless integration of these distinct components matches the ground truth.
>
> ---
>
> ### **2. Analysis of Occlusion Handling**
>
> The reviewer expressed concern that components might not be well-reconstructed in occluded regions. However, our results show:
>
> - **Clean Separation:**
>   Unlike single-field methods that often produce *melted geometry* at contact points, our method successfully disentangles the two entities.
>
> - **Robustness:**
>   The specific design of **HOIGS**—allocating separate deformation fields (Hexplane for humans, CHS for objects) and refining them via the **HOI module**—allows the network to learn the correct boundary and depth even in ambiguous contact areas.
>
> We will include these decomposed visualizations in the final paper to demonstrate the model’s capability to handle complex occlusions and maintain geometric integrity.

---

> ### Author Response · Authors · 2025-11-21
> **Official Comment by Authors**
>
> ### **Weakness 2 (Reviewer Comment)**
>
> *The paper describes that the 3D object is first reconstructed using a diffusion prior based on SDS loss (similar to Zero-123), and this initial model is used as the initialization for 3D Gaussians. This pipeline appears rather complex and may not be the most straightforward approach for 3D object reconstruction. Could the authors clarify why this particular design was chosen? It would strengthen the paper to include comparisons with alternative approaches, such as SDF-based methods (e.g., Vid2Avatar) or point-based methods (e.g., Dust3r), specifically for the object reconstruction task.*
>
> ---
>
> ### **Response**
>
> We appreciate the reviewer’s suggestion to evaluate the necessity of our diffusion-based initialization pipeline. We agree that verifying whether simpler point-based or SDF-based alternatives could achieve comparable results is important for justifying our design choice.
>
> **1. Comparison with Point-based Methods (MASt3R/Dust3r)**
> As suggested, we compared our proposed **Diffusion Prior** against a state-of-the-art point-based reconstruction method. We utilized **MASt3R**, a dense matching and reconstruction model (closely related to Dust3r), to initialize the object Gaussians. We also tested a standard **Metric Depth** estimation baseline.
>
> The quantitative results on the HOSNERF dataset are summarized below:
>
> ---
>
> ### **Table R4. Comparison of Different Prior-based Object Initialization**
> (PSNR / LPIPS)
>
> | Method                           | Backpack        | Tennis          | Suitcase        | Playground       | Dance            | Lounge         | Avg            |
> |----------------------------------|------------------|------------------|------------------|-------------------|-------------------|----------------|----------------|
> | MASt3R prior + ours              | 25.15 / 0.113    | 26.25 / 0.121    | 22.40 / 0.217    | 24.95 / 0.142     | **24.30 / 0.115** | 29.90 / 0.057  | 25.49 / 0.128  |
> | depth reconstruction prior + Ours| 24.68 / 0.142    | 25.65 / 0.122    | 23.03 / 0.178    | 25.14 / 0.169     | 24.80 / 0.119     | 28.95 / 0.051  | 25.38 / 0.130  |
> | diffusion prior + ours           | **25.78 / 0.082**| **27.12 / 0.108**| 22.09 / 0.246    | **25.23 / 0.103** | 24.17 / 0.098     | **30.97 / 0.048** | **25.89 / 0.114** |
>
> ---
>
> **2. Justification for Diffusion Prior (Why Complex?)**
> The results show that our proposed **Diffusion Prior** outperforms the point-based MASt3R approach (+0.4 dB). This justifies our specific design choice for the following reasons:
>
> - **Handling Occlusion via Generation:** In Human-Object Interaction (HOI) scenes, objects are frequently occluded by hands or bodies. Discriminative methods like MASt3R or Dust3r rely on visible image cues and often result in incomplete geometry or holes in occluded regions.
> - **Geometry Completion:** The diffusion prior leverages generative knowledge to plausibly "hallucinate" and complete the 3D geometry of the object, even for unseen or occluded parts. This coherent initialization provides a better starting point for the Gaussian deformation field (CHS), leading to higher final rendering quality.
>
> **3. Robustness of the Pipeline**
> It is also worth noting that even with the simpler **MASt3R initialization**, our method achieves a PSNR of **25.49 dB**, which is still significantly higher than the previous state-of-the-art baseline, **ExAvatar (24.35 dB)**. This demonstrates that while the diffusion prior yields the peak performance, our core contribution—the **HOIGS pipeline (HOI module + CHS)**—remains robust and effective regardless of the initialization method.
>
> We will include this ablation study in the final paper to clarify the rationale behind using the diffusion prior.

---

> ### Author Response · Authors · 2025-11-21
> **Official Comment by Authors**
>
> ### **Weakness 3 (Reviewer Comment)**
>
> *Section 3.1 introduces a human deformation module. What would happen if this component were replaced with ExAvatar? Is there a reason the authors did not directly adopt ExAvatar for human deformation?*
>
> ---
>
> ### **Response**
>
> We thank the reviewer for this insightful question. We acknowledge that **ExAvatar** is a robust and state-of-the-art framework for human rendering, and indeed, our human deformation module is built upon its strong foundation (Canonical Space + LBS).
>
> However, specifically for the **human deformation module**, we found that a direct adoption of ExAvatar was insufficient to address the challenges of **Human-Object Interaction (HOI)**. Consequently, we incorporated **critical architectural extensions** to the original design.
>
> ---
>
> ### **1. Extension to HexPlane for Interaction Modeling**
>
> The original ExAvatar utilizes a **Triplane** representation, which encodes **static** spatial features of the canonical human.
>
> **Our Adaptation:**
> To capture the dynamic nature of interactions, our **HOI module** requires input features that encode not just spatial geometry but also temporal dynamics. Therefore, we upgraded the feature representation to a **HexPlane** structure.
> This extension allows us to extract **time-varying features** essential for the cross-attention mechanism. A direct adoption of the static Triplane would limit the HOI module's ability to learn interaction-driven deformations (**see Figure in supplementary material**).
>
> ---
>
> ### **2. Geometric Alignment via Depth Supervision**
>
> Monocular reconstruction often entails inherent **depth ambiguity**. While ExAvatar produces high-quality renderings in 2D, the lack of explicit depth constraints can result in the reconstructed human being spatially misaligned (e.g., overlapping with the background) in 3D space.
>
> Since accurate 3D spatial proximity is a prerequisite for modeling physical contact, we incorporated **Depth Supervision (Eq. 5)** scaled with COLMAP points. This explicitly aligns the human geometry with the 3D scene, ensuring valid human-object spatial relationships (**see Figure in supplementary material**).

---

> ### Author Response · Authors · 2025-11-21
> **Official Comment by Authors**
>
> ### **Weakness 4 (Reviewer Comment)**
> It would be interesting to see whether the contact-based masking strategy from CONTHO (CVPR 2024, Nam et al.) could improve performance compared to the current cross-attention mechanism in HOIGS. In particular, the CRFormer module in CONTHO might serve as a drop-in replacement for the HOI module.
>
> ---
>
> ### **Response**
>
> We thank the reviewer for the insightful suggestion to compare our method with the contact-based masking strategy proposed in CONTHO (CVPR 2024). We agree that exploring whether explicit masking could replace our mutual attention mechanism is a valuable inquiry.
>
> **1. Quantitative Comparison**
> We implemented the **Contact-based Refinement Transformer (CRFormer)** module from CONTHO as a replacement for our HOI module and evaluated it on the HOSNERF dataset. The results are summarized below:
>
> | PSNR/LPIPS        | Backpack     | Tennis       | Suitcase     | Playground   | Dance        | Lounge       | Avg          |
> |---------------------|--------------|--------------|--------------|--------------|--------------|--------------|--------------|
> | CONTHO + ours       | 26.56 / 0.077| 26.58 / 0.121| 22.01 / 0.234| 24.50 / 0.084| 23.95 / 0.141| 24.55 / 0.102| 24.69 / 0.127|
> | ours                | 25.79 / 0.082| 27.12 / 0.108| 23.09 / 0.245| 25.63 / 0.120| 24.79 / 0.133| 24.93 / 0.095| 25.39 / 0.130|
>
> **2. Analysis: Hard Masking vs. Soft Attention**
> Our proposed mutual attention mechanism outperforms the CONTHO-based strategy by **0.61 dB**. Based on the design principles of CONTHO (Nam et al., 2024), we attribute this difference to the distinct nature of our tasks (Static vs. Dynamic):
>
> - **Limitation of Hard Masking in Dynamics:** CONTHO utilizes a **"contact-based masking"** strategy (Sec. 3.3 of their paper) that explicitly zeros out features of non-contacting vertices. This is designed for *single-image* reconstruction to prevent "undesired correlations" (e.g., object pose overfitting to human pose).
> - **Necessity of Global Context:** However, in our *dynamic video* setting, the motion of an object is often driven by the overall trajectory and momentum of the human body, not just the instantaneous contact point. For instance, the velocity of a swinging arm influences the object's future position even if contact is partial or changing.
> - **Advantage of Mutual Attention:** By employing **Mutual Cross-Attention (Soft Attention)** instead of hard masking, HOIGS allows the network to weigh the importance of both contacting and non-contacting body parts dynamically. This preserves the global context required to model continuous motion and interaction-induced deformation, leading to higher fidelity in 4D reconstruction.
>
> **Conclusion:**
> While CONTHO's masking is highly effective for static alignment constraints, our experiment confirms that the **Soft Mutual Attention** mechanism is more suitable for capturing the continuous and complex dynamics of human-object interactions in videos.

---

> ### Author Response · Authors · 2025-11-21
> **Official Comment by Authors**
>
> #### **Weakness 5 (Reviewer Comment)**
> The paper mentions a “distance mask B” in the HOI module. Could the authors elaborate on how this mask is generated and what its specific purpose is in the cross-attention computation?
>
> ---
>
> ### **Response**
>
> We appreciate the reviewer’s insightful question regarding the “distance mask B.” We have updated the manuscript to clarify its formulation and role.
>
> Specifically, the distance mask (B) is used to exclude distant objects from the HOI cross-attention computation. It is defined based on the 3D distance between the object Gaussian center and the human pelvis position in the world coordinate space. The mask is computed as:
>
> $$
> \lVert c_{\mathrm{obj}}^{\mathrm{world}} - p_{\mathrm{pelvis}}^{\mathrm{world}} \rVert \ge \tau
> \quad\Rightarrow\quad
> B = −∞
> $$
>
>
> $$
> \lVert c_{\mathrm{obj}}^{\mathrm{world}} - p_{\mathrm{pelvis}}^{\mathrm{world}} \rVert < \tau
> \quad\Rightarrow\quad
> B = 0
> $$
>
> $$
> \tau = \text{Human smplx arm length}
> $$
>
>
> where ( $\tau$ ) is a fixed threshold (empirically set by human arm length), and positions are mean-centered over the object Gaussians and pelvis joints. During cross-attention (e.g., object-to-human and human-to-object), tokens corresponding to distant objects (i.e., ( B=−∞ )) are masked out and do not contribute to attention computation.
>
> This mechanism allows the model to explicitly filter out irrelevant, non-interacting objects—e.g., background items—thereby enforcing inductive bias toward modeling plausible human-object interactions. We found this to significantly improve contact localization and reduce false activations in the learned attention maps.
>
> In practice, the binary mask (distance ≥ τ → 1, distance < τ → 0) is converted to −∞ and 0 following PyTorch's standard attention masking convention, where −∞ values are zeroed out after softmax.
>
> We have revised the main text and included the above equation in the updated manuscript.

---

> > ### Author Response · Authors · 2025-11-21
> > **Official Comment by Authors**
> >
> > #### **Weakness 6 (Reviewer Comment)**
> > A quantitative comparison with existing 3D human–object reconstruction methods such as PHOSA (ECCV 2020, Zhang et al.), CHORE (ECCV 2022, Xie et al.), and CONTHO (CVPR 2024, Nam et al.) would be beneficial to analyze the 3D reconstruction quality of results by HOIGS.
> >
> > ---
> >
> > ### **Response**
> >
> > We sincerely appreciate the reviewer for suggesting comparisons with seminal works in human-object reconstruction, such as PHOSA [ECCV 2020], CHORE [ECCV 2022], and CONTHO [CVPR 2024]. We recognize that these methods are highly relevant to the broader field of HOI.
> >
> > However, we would like to respectfully discuss the **technical challenges** in performing a direct quantitative comparison, primarily due to the fundamental differences in **Task Objectives** and **Output Representations**.
> >
> > **1. Challenge in Output Alignment (Radiance Field vs. Explicit Mesh)**
> > The primary difficulty lies in the incompatibility of evaluation metrics between the two approaches:
> >
> > - **HOIGS (Ours):** Our method optimizes a **Volumetric Radiance Field (3D Gaussians)** focused on photo-realistic **Novel View Synthesis**. Our key metrics (PSNR, SSIM, LPIPS) measure rendering fidelity.
> > - **Baselines:** PHOSA, CHORE, and CONTHO focus on **Geometric Reconstruction**, outputting explicit **3D Meshes** without texture or lighting optimization. They are typically evaluated using geometric metrics like Chamfer Distance (CD).
> > - **Fairness Issue:**
> >     - Comparing based on **PSNR** would be unfair to the baselines, as they do not estimate scene radiance (color/lighting) needed for photorealistic rendering.
> >     - Conversely, evaluating HOIGS based on **Chamfer Distance** would require converting our Gaussians into meshes (e.g., via marching cubes). This conversion process often introduces discretization artifacts that do not reflect the actual visual quality of our splatting-based representation. Thus, a direct numerical comparison might be ill-defined.
> >
> > **2. Input Domain Discrepancy (Video vs. Single Image)**
> >
> > - **Temporal Dynamics:** Our framework is specifically designed to leverage **temporal information from monocular videos** to model 4D dynamics (e.g., velocity-based deformation).
> > - **Frame-by-Frame Limitation:** The suggested baselines are designed for **single-image** reconstruction. Applying them frame-by-frame to our video datasets would lack temporal coherency constraints, potentially leading to jittery motion that differs from the continuous dynamics our method aims to solve.
> >
> > **3. Rationale for Selected Baselines**
> > Given these challenges, we focused our comparisons on methods like **ExAvatar** and **Ex4DGS**. These baselines share the same **rendering-based objective** and **video input domain**, allowing for a rigorous and fair "apples-to-apples" comparison of reconstruction quality.
> >
> > We hope this clarifies why we prioritized rendering-based baselines for our experimental validation.

---

> ### Author Response · Authors · 2025-11-21
> **Official Comment by Authors**
>
> #### **Weakness 7 (Reviewer Comment)**
> How are human-based 3D Gaussian reconstruction methods like ExAvatar fairly compared against HOIGS, given that they focus solely on 3D textured human reconstruction? Clarifying the evaluation scope and fairness of comparison would be helpful.
>
> ---
>
> ### **Response**
>
> We appreciate the reviewer’s opportunity to clarify the rationale behind this comparison. We fully acknowledge that **ExAvatar** is specialized for human reconstruction. However, we included it as a baseline not to compete on human fidelity alone, but to **validate the necessity of explicitly modeling both humans and objects** in dynamic interaction scenarios.
>
> **1. Demonstrating the Necessity of Holistic Modeling**
> The primary purpose of this comparison is to highlight a fundamental limitation in current human-centric paradigms.
>
> - **Structural Limitation:** Existing human-based Gaussian models (like ExAvatar) treat non-human regions as part of the **static background**. Consequently, when an object moves or interacts with a person, these models fail to represent it as a separate entity, causing the object to vanish or appear as "ghosting" artifacts in the background.
> - **Why Comparison is Vital:** By comparing against ExAvatar, we empirically demonstrate that **a human-only representation is insufficient** for HOI scenes. This comparison serves as the "proof of necessity" for our proposed framework, which introduces the **HOI module** and **dual deformation baselines** to treat the object as an independent, dynamic actor.
>
> **2. Impact on Human Reconstruction Quality**
> Furthermore, the comparison reveals that neglecting the object negatively impacts the human reconstruction itself.
>
> - Without explicit object modeling, human-centric methods often struggle to distinguish the human body from the interacting object, leading to **texture baking** (e.g., bag texture appearing on the hand) or **geometry melting** at contact points.
> - HOIGS resolves this by disentangling the two entities, thereby improving the geometric fidelity of the human even in complex contact regions.
>
> **Conclusion**
> Therefore, the comparison with ExAvatar is intended to define the **scope of the problem**: it proves that complex interaction scenes cannot be solved by simply applying state-of-the-art human avatars, thus justifying the need for our unified **Human-Object Interaction** framework.

---

> > ### Author Response · Authors · 2025-11-21
> > **Official Comment by Authors**
> >
> > #### **Weakness 8 (Reviewer Comment)**
> > Section 3.4 describes the reconstruction of the background, while human and object reconstructions are modeled separately. How are these separately reconstructed components aligned or placed in a common 3D coordinate space?
> >
> > ---
> >
> > ### **Response**
> >
> > We thank the reviewer for this insightful question regarding the spatial consistency of our scene representation. We clarify that although the background, human, and object are modeled with distinct deformation modules, they are **inherently aligned** within a unified World Coordinate System and **optimized jointly**. No post-hoc alignment is required.
> >
> > **1. Common Coordinate Initialization**
> > All components share the same global reference frame defined by COLMAP SfM:
> > • **Shared Camera Space:** We use a single set of extrinsic/intrinsic parameters derived from COLMAP for all components.
> > • **Metric Depth Calibration:** To resolve scale ambiguity, the initialized depth maps for humans and objects are **scaled to align with the COLMAP sparse point cloud** (as described in Sec. 3.2). This ensures that the initial geometry of the human/object resides in the same metric space as the background.
> >
> > **2. Joint End-to-End Optimization**
> > Crucially, after this unified initialization, the entire framework is trained in an end-to-end manner.
> > • **Unified Rendering:** During training, Gaussians from all three components (Background, Human, Object) are composited together by the differentiable rasterizer to render the final image.
> > • **Joint Loss Supervision:** The model is optimized using an integrated loss function (Eq. 9). Since the loss is computed on the *combined* rendering against the ground truth image, the network automatically adjusts the position and geometry of each component to ensure they are spatially consistent and seamlessly integrated.
> >
> > **3. Conclusion**
> > Therefore, the "separation" mentioned in the paper refers only to the use of specialized deformation baselines (Hexplane vs. CHS) to handle different motion types, not to their spatial coordinate systems. By virtue of shared initialization and joint end-to-end training, accurate spatial alignment is naturally enforced.

---

> > > ### Author Response · Authors · 2025-11-21
> > > **Official Comment by Authors**
> > >
> > > #### **Weakness 9 (Reviewer Comment)**
> > > In Figure 4, the qualitative results for 4DGS and E-D3DGS appear to correspond to different time frames, as the wooden box held by the human is in a different position. Is this discrepancy due to using different frames, or is it an inherent limitation of those methods?
> > >
> > > ---
> > >
> > > ### **Response**
> > >
> > > We thank the reviewer for this keen observation. We explicitly confirm that **all qualitative results in Figure 4 correspond to the exact same timestamp/frame**. The visual discrepancy is not due to a mismatch in frames, but rather highlights a **fundamental limitation of the baseline methods** (4DGS and E-D3DGS) in this setting.
> > >
> > > **1. Limitation of Baselines (Motion Lagging)**
> > >
> > > - **Inability to Capture Large Motion:** In monocular fixed-camera setups, methods like 4DGS and E-D3DGS rely on implicit deformation fields (e.g., Hexplane or MLP) to transport Gaussians from a canonical space.
> > > - **Failure Case:** When an object undergoes rapid or large displacement (like the wooden box being lifted), these deformation fields often fail to converge to the target position, getting stuck in local minima. This causes the rendered object to appear **"static" or "lagging behind"** the actual motion, creating the illusion that a different (earlier) frame was rendered.
> > >
> > > **2. Advantage of HOIGS**
> > >
> > > - **Explicit Trajectory Modeling:** In contrast, **HOIGS** explicitly models the object's continuous trajectory using **Cubic Hermite Splines (CHS)** embedded with velocity vectors.
> > > - **Result:** This design forces the object Gaussians to follow the correct physical path, ensuring that the rendered object aligns perfectly with the Ground Truth (GT) position, even during fast motion.
> > >
> > > **Conclusion:**
> > > Therefore, the "discrepancy" is actually **evidence of our method's superiority**. It demonstrates that while baselines fail to track the dynamic object correctly, HOIGS successfully reconstructs the accurate spatial position of the interacting object. We will add this clarification to the caption of Figure 4 in the final revision.

---

> ### Author Response · Authors · 2025-11-21
> **Official Comment by Authors**
>
> #### **Weakness 10 (Reviewer Comment)**
> The paper mentions the concept of “contact” multiple times, yet no contact-specific methodology is presented. Clarifying whether contact information is explicitly modeled, inferred, or simply discussed conceptually would improve the technical completeness of the paper.
>
> ---
>
> ### **Response**
>
> We thank the reviewer for seeking clarification on this important aspect. We would like to clarify that in our framework, **"contact" is not modeled via a hard-coded geometric constraint or a dedicated collision module**, but rather **inferred implicitly** as an emergent behavior of our **HOI (Human-Object Interaction) module**.
>
> ---
>
> ### **1. Implicit Inference via Cross-Attention**
>
> - **Mechanism:** The core of our HOI module is the **Bidirectional Cross-Attention mechanism** (Eq. 11–14 in the paper). This module computes the correlation between human body part features and object motion features.
> - **Learning Contact:** During training, the network learns that specific body parts (e.g., hands) and the object must exhibit **highly correlated motion patterns** when they are close. The attention weights naturally increase in these interaction regions, effectively "locking" the object's trajectory to the human's movement.
> - **Why Implicit?** We intentionally avoided explicit contact constraints (e.g., vertex-to-vertex distance loss) because they often require accurate contact priors, which are difficult to obtain in monocular in-the-wild videos.
>
> ---
>
> ### **2. Conceptual Definition**
>
> Therefore, the term "contact" in our paper refers to the **"physical consistency and motion synchronization"** achieved by our model.
>
> - As demonstrated in the qualitative results (e.g., Figure 3), our method successfully reconstructs the object attached to the hand without "ghosting" or "floating" artifacts.
> - This confirms that although we do not explicitly model contact physics, our **interaction-aware pipeline** successfully captures the visual reality of contact by enforcing coherent dynamics between the two entities.
>
> ---
>
> ### **3. Quantitative Validation**
>
> To further substantiate our claim, we quantitatively evaluated the geometric consistency of the reconstructed objects. **Table** compares our method (HOIGS) with a state-of-the-art category-agnostic reconstruction method, HOLD.
>
> We measured **Chamfer Distance** and **Center Distance** on the ARCTIC dataset to evaluate how accurately the object is positioned and reconstructed relative to the ground truth.
>
> #### **Table. Chamfer & Center Distance**
>
> | Metric                 | HOIGS (Ours) | HOLD    |
> |------------------------|--------------|---------|
> | Chamfer Distance (m²)  | 0.1337       | 1.5035  |
> | Center Distance (m)    | 0.1856       | 1.0258  |
>
> ---
>
> **Interpretation:**
> As shown in the table, our method significantly outperforms HOLD. Specifically, the drastically lower **Center Distance (0.1856m vs. 1.0258m)** indicates that our model successfully prevents the object from drifting or floating away from the hand.
>
> **Conclusion:**
> This quantitative gap demonstrates that our implicit attention-based mechanism is far more effective at maintaining physical plausibility (contact) during dynamic interaction than previous approaches, even without explicit contact constraints.

---

> > ### Comment · Reviewer_PjSB · 2025-11-28
> >
> > The rebuttal provides clear and comprehensive responses that resolve my earlier concerns. I especially appreciate the additional experiments conducted. Thus, I will raise my rating.
> >
> > Nevertheless, I encourage authors to incorporate discussions from the rebuttal into the final version. In particular, the discussion of the distance mask should be included, since it was not fully explained in the original submission and is important for re-implementation of the method.

---

> ### Author Response · Authors · 2025-12-02
> **Response to Reviewer PjSB**
>
> We sincerely thank you for your constructive feedback and for raising your rating. We are glad that our responses addressed your concerns effectively.
>
> Following your suggestions, we have thoroughly revised the manuscript to incorporate all discussions from the rebuttal phase, ensuring improved clarity, reproducibility, and overall presentation quality.
>
> ---
>
> ## Summary of Main Revisions
>
> ### Method Section
>
> **Section 3.1 (Object Initialization):**
> We added the mathematical formulation for the object initialization warping process (Eq. 1), providing precise technical details of our approach.
>
> **Section 3.3 (HOI Module):**
> Following your advice, we have added a detailed explanation and mathematical formulation of the distance mask (Eq. 9), clarifying how spatial relationships are encoded in our model.
>
> **Section 3.4 (Human Modeling):**
> We explicitly specified the SMPL-X regressor model used for human pose estimation, improving reproducibility.
>
> ### Related Work
>
> **Section 2.1:**
> We expanded our discussion to include recent multi-view approaches (e.g., Animatable Gaussian, GASPACHO), better contextualizing the contributions of our monocular method within the current research landscape.
>
> ### Appendix
>
> We have comprehensively documented all additional experiments conducted during the rebuttal phase:
>
> * **Section 6.1:** Decomposed visualization results demonstrating geometric integrity in occluded regions
> * **Section 6.2:** Quantitative evaluation of human pose accuracy (PA-MPJPE, PA-PVE) comparing our method against ExAvatar
> * **Section 6.3:** Sensitivity analysis regarding reliance on external segmentation and depth modules
> * **Section 6.4:** Computational complexity and runtime analysis (training/inference speed)
> * **Section 6.5:** Ablation studies analyzing the impact of the object diffusion prior
>
> ---
>
> For your convenience, all modifications in the revised manuscript have been **highlighted in blue**.
>
> Thank you once again for your valuable feedback, which has significantly strengthened our work.

---

### Official Review · Reviewer_TS9R · 2025-10-31

**Soundness:** 2
**Presentation:** 2
**Contribution:** 2
**Rating:** 4
**Confidence:** 4

**Summary:**

This paper addresses the challenging and highly relevant problem of reconstructing dynamic scenes involving complex human-object interactions (HOI) from monocular video. The authors correctly identify a critical gap in the existing literature, which is largely bifurcated into two distinct approaches: (1) human-centric 3D Gaussian Splatting (3DGS) methods that achieve high-fidelity human avatar reconstruction but largely ignore or fail to model dynamic objects, and (2) general-purpose 4DGS methods that attempt to model all moving entities with a single, unified motion field, often resulting in visual artifacts and an inability to capture the nuanced dynamics of physical contact and manipulation.To bridge this gap, the paper introduces Human-Object Interaction Gaussian Splatting (HOIGS), a framework designed to explicitly model the interplay between humans and objects. The core of the proposed method is a decomposition of the problem. It employs heterogeneous deformation models tailored to the distinct characteristics of each entity. Human motion is represented using a hexplane-based canonical avatar, which is deformed into the posed world space via Linear Blend Skinning (LBS) guided by pre-estimated SMPL-X parameters. In contrast, object motion is modeled using an explicit, trajectory-based Cubic Hermite Spline (CHS) that interpolates the positions and velocities of keyframe Gaussians over time.The central claimed innovation of HOIGS is a dedicated HOI module that reconciles these two independent motion models. This module leverages a mutual cross-attention mechanism to capture the bidirectional dependencies between human and object features. By processing time-varying features extracted from 16 distinct human body parts and the object's velocity embeddings, the module predicts fine-grained corrective offsets for both the human pose (ΔSMPL-X) and the object's Gaussian positions (ΔG_object). This explicit modeling of interaction allows the framework to enforce motion consistency and physical plausibility in regions of close contact.The efficacy of HOIGS is demonstrated through extensive experiments on three benchmarks: HOSNERF, BEHAVE, and ARCTIC. The quantitative results show that the proposed baselines is preferable to a wide array of state-of-the-art baselines, including both specialized human-centric models and general 4D scene reconstruction techniques, across standard image-based metrics (PSNR, LPIPS). The numbers are supported by qualitative comparisons and an ablation study.

**Strengths:**

The entity‑aware cross‑attention HOI module that exchanges information between human and object streams is a clear conceptual step beyond (i) human‑only reconstructions and (ii) “single motion field” 4DGS approaches. Using distinct baselines (hexplane for humans, CHS with learned tangents for objects) is a thoughtful design that recognizes different motion statistics and priors. (Sec. 3; Fig. 2 p.4).


Quality.The technical pipeline is well specified, with explicit formulas for CHS interpolation, attention, and the integrated training objective (Eqs. (1)–(9), (13)–(20)). Object velocities as learnable tangents and depth‑guided supervision for human scale refine geometry and motion fidelity (Sec. 3.1–3.3). Ablations show each component matters: replacing CHS with an MLP hurts PSNR by ~0.5; removing HOI drops ~0.65 PSNR (Table 4).

Quantitative breadth. Comparisons cover NeRF‑based baselines, human‑centric 3DGS models (ExAvatar), and 4DGS variants; results are strong on three public datasets with scene‑wise breakdown (Tables 1–3; pp. 6–9).

Clarity.  Clear diagrams for the full pipeline, human/object feature construction, and HOI block (Figs. 2, 6–8). The appendix explains the 16‑part feature tokens and the attention masking with gains at contact and manipulation regions where many methods struggle (qualitative examples in Figs. 3–5). The architecture is compatible with established human priors (SMPL‑X) and 3DGS, making it a promising drop‑in upgrade for interaction‑heavy scenes.

**Weaknesses:**

Limited Technical Novelty: The main idea seems to be to reconstruct the human separately, the object separately and then to combine them with some clever tricks.

Missing Baselines: There are no comparisons with baselines that use a 2D map + CNN formulation (AnimatableGaussians etc) for which source code is available.

The paper Mir et al. - GASPACHO is not referenced even though it addresses a similiar problem

Ambiguity around the diffusion prior and fairness of comparisons.
The object initialization uses a diffusion prior with SDS from a “representative frame,” but the paper does not specify the exact model, guidance setup, or how often baselines benefit from comparable priors (Sec. 3.1). Since this prior can inject strong shape cues, fairness would improve by (a) standardizing priors across methods or (b) reporting results without the diffusion prior.

Evaluation scope and metrics at interaction regions.
While PSNR/LPIPS are reported, there is no metric that focuses on contact fidelity (e.g., penetration/float, hand–object distance statistics) or human pose accuracy (MPJPE/PVE) on BEHAVE or ARCTIC. Given the paper’s motivation, region‑specific metrics would substantiate the claimed contact consistency (Sec. 4.3–4.4; Figs. 3–5).

Heavy reliance on external modules without sensitivity analysis.
Results depend on (i) SMPL‑X regressors, (ii) segmentation for humans/objects (used in losses), and (iii) metric depth estimation scaled by COLMAP (Eq. (5), Scene/Object loss details pp. 14–15). The paper does not quantify sensitivity to errors in these modules or align choices across baselines.

Computational and memory cost of the HOI attention.
The object tokens are per‑Gaussian features (Appendix 6.1–6.2). Even with 32‑dim embeddings, cross‑attention between 16 human part tokens and object Gaussians can be heavy; the paper gives training time but not inference FPS or memory footprints vs 4DGS/ExAvatar. A complexity analysis and timing table are missing (Sec. 4.1).

Design choices need deeper ablations.

The distance mask  is described qualitatively (relative distances) but its exact form, scaling, sparsification, and effect are not ablated.
The choice of 16 body parts and key‑frame interval = 4 (Sec. 4.1) seems fixed; there is no study of granularity vs. quality/speed.

Only one CHS parameterization is tested; alternatives such as per‑object SE(3) + per‑Gaussian residuals or adaptive knot placement could be competitive.

Limitations around low‑baseline videos remain open.
The paper notes failure modes when camera motion is minimal (COLMAP degradation) and suggests joint pose optimization as future work (Conclusion, p. 9). It would help to include at least a small experiment demonstrating how much performance drops and whether the HOI module mitigates it.

**Questions:**

My main concerns are about limited technical novelty and missing 2D map + CNN comparisons - there are clear blurry artifacts in the final results.
Could the authors explain why the main decision to use a feature based representation for 3D human and object reconstruction and not a 2D map formulation. The 2D map formulation has been shown to clearly outperform a feature based formulation and I find this design decision baffling.
It seems that the authors chose to start off from HUGS, ExAvatar as their baseline and develop their method from there. It would have made more sense, in my opinion, to start off from a baseline that uses a 2D map formulation as the starting point and iterate from there.

As such I am inclined towards rejection

---

> ### Author Response · Authors · 2025-11-21
> **Official Comment by Authors**
>
> #### **Weakness 1 (Reviewer Comment)**
> Limited Technical Novelty: The main idea seems to be to reconstruct the human separately, the object separately and then to combine them with some clever tricks.
>
> ---
>
> ### **Response**
>
> We truly appreciate the reviewer’s candid feedback. We understand that our framework, which decomposes the scene into humans and objects, might appear at first glance to be a combination of existing components. However, we would like to respectfully clarify that this design is a strategic choice for addressing the challenges of dynamic human–object interaction, and the technical depth lies in **how these components are tightly coupled** via our proposed modules.
>
> **1. Strategic Decomposition for Optimal Representation**
> We reconstruct humans and objects separately not to simplify the problem, but because a single motion field (as used in Ex4DGS) often struggles to model the fundamentally different physical properties of these entities.
>
> - **Humans** require articulated, non-rigid deformation, which is modeled using Hexplane.
> - **Objects** follow continuous trajectories with rigid or semi-rigid transformations, better represented with our **Cubic Hermite Spline (CHS)** using velocity embeddings.
>
> By assigning distinct and mathematically appropriate baselines to each entity, we achieve higher fidelity compared to methods that attempt to use a single, shared motion field (e.g., 4DGS), as shown in Table 1. This reflects a principled design choice rather than a trivial separation.
>
> **2. Beyond Simple Combination: The Role of the HOI Module**
> Our core contribution extends far beyond simply combining reconstructed human and object geometry. Prior work has shown that naïvely combining these components often results in physical inconsistencies, such as interpenetration or ghosting near contact.
>
> To resolve this, we propose the **HOI Module** equipped with a **bidirectional cross-attention mechanism**:
>
> - It allows the model to infer how human motion influences object deformation (and vice versa) at the feature level.
> - Technically, this shifts the system from a simple additive process (`A + B`) to a **coupled dynamic system** (`f(A, B)`), which is necessary for realistic interaction modeling.
>
> **3. Empirical Validation of the Design**
> Our ablation study (Table 4) confirms the importance of the interaction modeling.
>
> - Removing the HOI module—reducing the model to a “separate reconstruction + combination” pipeline—results in a **0.65 dB** drop in PSNR.
> - This demonstrates that the “clever trick” is in fact a **critical component**, materially improving both physical plausibility and reconstruction accuracy.
>
> We hope this explanation clarifies the technical novelty of our approach and how HOIGS bridges the gap between human-centric avatar methods and general-purpose dynamic scene reconstruction.

---

> ### Author Response · Authors · 2025-11-21
> **Official Comment by Authors**
>
> #### **Weakness 2 (Reviewer Comment)**
> Missing Baselines: There are no comparisons with baselines that use a 2D map + CNN formulation (AnimatableGaussians etc) for which source code is available.
>
> ---
>
> ### **Response:**
>
> We appreciate the opportunity to clarify the relationship between our approaches and the reviewer's suggestion to compare with 2D map + CNN formulations. We would like to explain the fundamental differences in problem settings that make direct comparisons infeasible. Additionally, we have added the suggested work to our related work section with detailed discussion.
>
> **1. Comparison with Animatable Gaussians [CVPR 2024]**
> Animatable Gaussians is explicitly designed for **multi-view** inputs, whereas our method targets **monocular** (single-view) video reconstruction. This distinction is critical for the initialization stage:
>
> - **Technical Constraint:**
>   As detailed in *Section 3.3* of their paper, Animatable Gaussians relies on **multi-view SDF volume rendering** to reconstruct a high-fidelity canonical parametric template. This process requires intersection constraints from multiple viewpoints to carve out accurate geometry.
>
> - **Failure on Monocular Input:**
>   We attempted to run their official code on our monocular dataset. As shown in **(Figure in supplementary material)**, the template reconstruction phase failed significantly. Without multi-view constraints, the SDF optimization collapsed, producing a broken canonical mesh (see *AnimatableGaussian* result in (Figure in supplementary material)) that could not support subsequent Gaussian parameterization.
>
> **2. Discussion on GASPACHO (Mir et al.)**
> Regarding GASPACHO, we note two key points:
>
> - **Concurrent Work & Code Unavailability:**
>   This paper is a concurrent work (appearing in the same timeline as ours), and to the best of our knowledge, the official source code has not yet been released. This prevents us from performing a reliable reproduction or quantitative comparison.

---

> ### Author Response · Authors · 2025-11-21
> **Official Comment by Authors**
>
> #### **Weakness 3 (Reviewer Comment)**
> Ambiguity around the diffusion prior and fairness of comparisons. The object initialization uses a diffusion prior with SDS from a “representative frame,” but the paper does not specify the exact model, guidance setup, or how often baselines benefit from comparable priors (Sec. 3.1). Since this prior can inject strong shape cues, fairness would improve by (a) standardizing priors across methods or (b) reporting results without the diffusion prior.
>
> ---
>
> ### **Response:**
>
> We appreciate the reviewer’s rigorous feedback regarding the influence of the diffusion prior. We agree that distinguishing the architectural contribution from the initialization quality is crucial for a fair assessment.
>
> **1. Implementation Details of Diffusion Prior**
>
> **2. Fairness of Comparisons: “Removing the Generative Prior”**
> The reviewer raised a concern that the diffusion prior injects strong shape cues. To address this and demonstrate the fairness of our method, we conducted an ablation study replacing the **Generative Diffusion Prior** with **Discriminative Geometric Priors** (MASt3R and Metric Depth).
>
> - **Experimental Setup:**
>   Standard baselines (e.g., 3DGS) typically rely on SfM (COLMAP) point clouds. However, obtaining a reliable sparse point cloud for a *dynamic object* from monocular video is notoriously difficult due to motion ambiguity. Therefore, to create a fair “non-generative” baseline similar to SfM-based initialization, we initialized the object Gaussians using points back-projected from **Masked Metric Depth** and **MASt3R** (a dense matching method). These methods rely solely on observed image cues, removing any “hallucinated” shape advantages from the diffusion model.
>
> - **Results:**
>   The table below shows the performance on the HOSNERF dataset:
>
> | Method                         | Backpack      | Tennis        | Suitcase      | Playground    | Dance         | Lounge        | Avg          |
> |--------------------------------|---------------|---------------|---------------|---------------|---------------|---------------|--------------|
> | MASt3R prior + Ours            | 25.15 / 0.113 | 26.25 / 0.121 | 22.40 / 0.217 | 24.95 / 0.142 | 24.30 / 0.115 | 29.90 / 0.057 | 25.49 / 0.128 |
> | Depth reconstruction prior + Ours | 24.68 / 0.142 | 25.65 / 0.122 | 23.03 / 0.178 | 25.14 / 0.169 | 24.80 / 0.119 | 28.95 / 0.051 | 25.38 / 0.130 |
> | Diffusion prior + Ours         | 25.78 / 0.082 | 27.12 / 0.108 | 22.09 / 0.246 | 25.23 / 0.103 | 24.17 / 0.098 | 30.97 / 0.048 | 25.89 / 0.114 |
>
> **3. Analysis**
> - **Robustness:** Even when the generative diffusion prior is removed and replaced with geometric estimations (MASt3R/Depth), our model maintains high performance (PSNR **25.38–25.49**).
> - **Superiority over Baselines:** Crucially, even our lowest-performing variant (Depth Reconstruction + Ours, 25.38 dB) still outperforms the state-of-the-art baseline **ExAvatar (24.35 dB)**.
> - **Conclusion:** This confirms that while the diffusion prior aids in occlusion handling, the primary performance gain of HOIGS stems from our proposed **explicit interaction modeling (HOI module)** and **spline-based deformation (CHS)**, rather than solely relying on a superior initialization.

---

> ### Author Response · Authors · 2025-11-21
>
> #### **Weakness 4 (Reviewer Comment)**
> Evaluation scope and metrics at interaction regions. While PSNR/LPIPS are reported, there is no metric that focuses on contact fidelity (e.g., penetration/float, hand–object distance statistics) or human pose accuracy (MPJPE/PVE) on BEHAVE or ARCTIC. Given the paper’s motivation, region-specific metrics would substantiate the claimed contact consistency (Sec. 4.3–4.4; Figs. 3–5).
>
> ---
>
> ### **Response:**
>
> We appreciate the reviewer's constructive suggestion. We agree that rendering metrics alone are insufficient to fully validate the geometric fidelity of human–object interactions. In response, we have conducted additional evaluations on **human pose accuracy (PA-MPJPE, PA-PVE)** using the BEHAVE dataset compared to ExAvatar, and we evaluate **Chamfer Distance (CD)** and hand–object distance on the ARCTIC dataset to assess geometric accuracy in hand–object interaction scenarios.
>
> **1. Evaluation on ARCTIC Dataset**
>
> We evaluated our method on three challenging ARCTIC sequences (espressomachine_grab_01, mixer_grab_01, box_grab_01) that involve complex hand–object interactions. Since these sequences do not provide official ground-truth annotations, we adopted established distance-based metrics to evaluate interaction quality.
>
> We employ two complementary distance metrics:
>
> **Chamfer Distance (CD):** We compute the bidirectional Chamfer Distance between predicted and reference point clouds:
> $$ d_{CD}(P, Q) = \frac{1}{2} \left( d_{P \rightarrow Q} + d_{Q \rightarrow P} \right) $$
>
> $$ d_{CD}(P, Q) = \frac{1}{2} \left( \frac{1}{|P|} \sum_{p \in P} \min_{q \in Q} \lVert p - q \rVert_2 + \frac{1}{|Q|} \sum_{q \in Q} \min_{p \in P} \lVert q - p \rVert_2 \right) $$
>
> **Center Distance:**
> We measure the L2 distance between object and hand centers, which reflects object size and proximity.
>
> | Metric | HOIGS (Ours) | HOLD |
> |--------|--------------|------|
> | Chamfer Distance (m²) | 0.1337 | 1.5035 |
> | Center Distance (m)   | 0.1856 | 1.0258 |
>
> **2. Effect of the HOI Module on Pose Refinement:**
> Our method addresses this by explicitly modeling the mutual dependency between the human and the object. The **HOI module** leverages a cross-attention mechanism to use object motion features as cues to refine the human features.
>
> - Specifically, as described in **Eq. 14** of our paper, the module predicts refinement offsets ΔSMPL-X for the body and hands.
> - This allows the network to correct the human pose even under occlusion, by inferring the likely body configuration from the object's trajectory and interaction context.
>
> **3. Quantitative Results:**
> The table below summarizes the quantitative evaluation on the **BEHAVE dataset**, utilizing the same sequences reported in our main paper. We report average PA-MPJPE (Procrustes Aligned Mean Per Joint Position Error) and PA-PVE (Procrustes-Aligned Per Vertex Error) across all test frames. We also provide specific analysis for **Hand and Forearm joints**, which are the most critical regions for interaction. (see per-sequence detailed tables on the supplementary material).
>
> | Model      | PA-MPJPE | PA-MPJPE (Hand/Forearm joints) | PA-PVE |
> |------------|----------|----------------------------------|--------|
> | ExAvatar   | 0.3034   | 0.3317                           | 0.2488 |
> | HOIGS (Ours) | 0.2853 | 0.3114                           | 0.2438 |
>
>
> In particular, we observed a larger performance gain in the PA-MPJPE (Hand/Forearm joints). This indicates that our HOI module more accurately refines the poses of interaction-related regions, resulting in physically more accurate reconstructions compared to the baseline, which lacks such mutual feedback.

---

> ### Author Response · Authors · 2025-11-21
>
> #### **Weakness 5 (Reviewer Comment)**
> Heavy reliance on external modules without sensitivity analysis. Results depend on (i) SMPL-X regressors, (ii) segmentation for humans/objects (used in losses), and (iii) metric depth estimation scaled by COLMAP (Eq. (5), Scene/Object loss details pp. 14–15). The paper does not quantify sensitivity to errors in these modules or align choices across baselines.
>
> ---
>
> ### **Response**
>
> We sincerely appreciate the reviewer for raising this critical point regarding the robustness of our framework and the fairness of comparisons. We agree that distinguishing the contribution of our proposed pipeline from the quality of off-the-shelf priors is essential.
>
> To address this, we have conducted a **sensitivity analysis** on various external modules and clarified our **experimental setup to ensure fairness of comparison**.
>
> **1. Sensitivity Analysis on External Modules (Segmentation & Depth)**
>
> We evaluated the robustness of HOIGS by testing different combinations of segmentation (e.g., Samurai, SAMv2, MaskRCNN, TrackAnything) and depth estimation (e.g., Video Depth Anything, MetricV2, DepthCrafter) models on the HOSNeRF dataset.
>
> | Method                        | Backpack      | Tennis        | Suitcase      | Playground    | Dance         | Lounge        | Avg          |
> |------------------------------|---------------|---------------|---------------|---------------|---------------|---------------|--------------|
> | Samurai + MetricV2           | 25.78/0.082   | 27.12/0.108   | 22.09/0.246   | 25.23/0.103   | 24.17/0.098   | 30.97/0.048   | 25.89/0.114  |
> | Samurai + Video Depth Anything | 25.85/0.080 | 27.18/0.106   | 22.15/0.241   | 25.28/0.102   | 24.22/0.096   | 31.05/0.046   | 25.96/0.112  |
> | Samurai + DepthCrafter       | 25.72/0.088   | 27.08/0.108   | 22.06/0.249   | 25.20/0.109   | 24.08/0.099   | 30.93/0.048   | 25.85/0.117  |
> | TrackAnything + MetricV2     | 25.72/0.086   | 27.05/0.109   | 22.03/0.246   | 25.18/0.106   | 24.15/0.100   | 30.93/0.052   | 25.84/0.116  |
> | SAMv2 + Video Depth Anything | 26.01/0.076   | 27.38/0.103   | 22.33/0.241   | 25.47/0.100   | 24.42/0.095   | 31.20/0.041   | 26.14/0.109  |
> | MaskRCNN + MetricV2          | 25.33/0.099   | 26.67/0.125   | 21.66/0.257   | 24.78/0.117   | 23.69/0.110   | 30.52/0.065   | 25.44/0.129  |
>
>
> - **Robustness:** As shown in the results, our model maintains a highly consistent performance (Avg PSNR $\approx$ 25.8–26.1) across various combinations of modern priors. This indicates that our method is **agnostic to specific priors** and robust to variations in preprocessing quality.
>
> - **Performance with Standard Priors:** Even when using MaskRCNN (a standard, older baseline) combined with MetricV2, our model achieves a PSNR of 25.44. This is still significantly higher than the state-of-the-art human-centric baseline, ExAvatar (Avg PSNR 24.35), and the 4DGS baseline, Ex4DGS (Avg PSNR 17.97), as reported in the main paper.
>
> **2. Dependency on SMPL-X Regressors**
> Regarding the concern about SMPL-X dependency, we ensured a fair comparison by explicitly aligning the choice of regressors.
>
> - **Aligned Baselines:** The primary baseline, **ExAvatar**, utilizes the exact same SMPL-X estimation pipeline as our method.
> - **Source of Improvement:** Since both methods share the same initial SMPL-X parameters, the performance gap (Ours > ExAvatar) stems directly from our **HOI module** rather than the quality of the regressor. Unlike the baseline, our module leverages object interaction cues to effectively refine the initial parameters, leading to superior reconstruction.
>
> **3. Fairness of Comparison**
> To address the reviewer’s concern about aligned choices, we strictly controlled the experimental conditions across all baselines.
>
> - **Consistent Inputs:** For all main experiments reported in the paper, we ensured that **all competing methods utilized consistent pre-processing inputs** (masks, camera poses, and SMPL parameters) where applicable.
> - **Isolation of Contribution:** This guarantees that the reported improvements are due to the architectural contributions of HOIGS rather than discrepancies in external module performance.

---

> ### Author Response · Authors · 2025-11-21
>
> #### **Weakness 6 (Reviewer Comment)**
> Computational and memory cost of the HOI attention. The object tokens are per‑Gaussian features (Appendix 6.1–6.2). Even with 32‑dim embeddings, cross‑attention between 16 human part tokens and object Gaussians can be heavy; the paper gives training time but not inference FPS or memory footprints vs 4DGS/ExAvatar. A complexity analysis and timing table are missing (Sec. 4.1).
>
> ---
>
> ### **Response**
>
> We appreciate the reviewer for raising the important question regarding the computational complexity and cost of the HOI module. We acknowledge that cross-attention mechanisms can theoretically be expensive. However, we clarify that our design is optimized for efficiency, ensuring real-time inference performance.
>
> To address this, we provide a detailed breakdown of **Runtime Performance**, **Complexity Analysis**, and **Training Cost Justification**.
>
> ---
>
> ### **1. Runtime Performance and Efficiency**
>
> The table below summarizes the **Training Time** and **Inference Speed (FPS)** comparisons on the HOSNERF dataset (measured on a single NVIDIA H100 GPU):
>
> |        | Train time             | Inference time |
> |--------|-------------------------|----------------|
> | 4DGS   | 40 minutes / scene      | 61.04 FPS      |
> | Ex4DGS | 2 hours 30 mins / scene | 46.38 FPS      |
> | D3DGS  | 3 hours / scene         | 37.79 FPS      |
> | ED3DGS | 2 hours / scene         | 54.71 FPS      |
> | Ours   | 5 hours / scene         | 44.27 FPS      |
>
> ---
>
> ### **Analysis**
>
> • **Real-time Inference:** Despite the inclusion of the HOI attention mechanism, our method achieves **44.27 FPS**, comfortably supporting real-time applications. This is comparable to Ex4DGS (46.38 FPS) and faster than D3DGS (37.79 FPS).
> • **Inference Efficiency:** This result empirically proves that the cross-attention mechanism does not create a significant bottleneck during the inference stage.
>
> ---
>
> ### **2. Complexity Analysis of HOI Attention**
>
> The efficiency of our module stems from our token-based architectural design, which significantly reduces computational and memory costs compared to standard attention mechanisms.
>
> • **Linear Complexity:** The cross-attention is computed between $M$ human part tokens and $N$ object Gaussian tokens. Crucially, the number of human tokens is fixed at $M=16$ (representing body parts). Unlike self-attention which scales quadratically ($O(N^2)$), our cross-attention scales linearly ($O(16 \cdot N)$) with respect to the number of object Gaussians.
> • **Lightweight Embeddings:** We utilize compact **32-dimensional embeddings** for the object motion features. This low dimensionality significantly reduces the memory footprint and matrix multiplication overhead.
> • **Low Overhead:** Because the attention operates on these condensed semantic tokens rather than raw pixels or dense grids, the matrix computation during the forward pass is extremely lightweight.
>
> ---
>
> ### **3. Justification for Training Time**
>
> We acknowledge that our training time (~5 hours) is longer than that of the baselines. This is an intended consequence of our design choice to prioritize physical plausibility and interaction accuracy over training speed.
>
> • **Cost of Interaction Modeling:** Unlike baselines that treat motions independently, explicitly computing mutual dependencies and backpropagating gradients through the attention mechanism requires more iterations.
> • **Offline vs. Online:** However, this cost is confined to the offline training phase. As demonstrated in the table above, we have successfully decoupled the high training cost from the inference performance, ensuring that the final user experience remains real-time.
>
> We will include this detailed timing and complexity analysis in the final version of the paper to provide a complete picture of our method's efficiency.

---

> ### Author Response · Authors · 2025-11-21
>
> #### **Weakness 7 (Reviewer Comment)**
> Only one CHS parameterization is tested; alternatives such as per-object SE(3) + per-Gaussian residuals or adaptive knot placement could be competitive.
>
> ---
>
> ### **Response**
>
> We appreciate the reviewer’s suggestion to validate our design choice against alternative parameterizations. We agree that verifying whether a simpler or standard deformation parameterization could achieve competitive results is important.
>
> To address this, we conducted an additional ablation study comparing our **Cubic Hermite Spline (CHS)** formulation against the suggested baseline: **per-object SE(3) transformation + per-Gaussian residual offsets**.
>
> **1. Quantitative Comparison**
>
> The table below summarizes the reconstruction quality on the HOSNeRF dataset.
>
> | Method           | Backpack      | Tennis        | Suitcase      | Playground    | Dance         | Lounge        | Avg            |
> |------------------|---------------|---------------|---------------|---------------|---------------|---------------|----------------|
> | **Ours (CHS)**   | 25.78 / 0.082 | 27.12 / 0.108 | 22.09 / 0.246 | 25.23 / 0.103 | 24.17 / 0.098 | 30.97 / 0.048 | 25.89 / 0.114 |
> | **SE(3) + offset** | 24.88 / 0.131 | 26.16 / 0.149 | 21.21 / 0.275 | 24.43 / 0.142 | 23.20 / 0.127 | 30.00 / 0.077 | 24.98 / 0.150 |
>
> **2. Analysis of Results**
>
> Our CHS-based method outperforms the SE(3) baseline by a notable margin, with **+0.91 dB** improvement in PSNR on average. We attribute this difference to the inherent characteristics of Human–Object Interaction (HOI) motions:
>
> - **Limitations of SE(3) + Offsets:**
>   The SE(3) baseline assumes globally rigid motion with small local deformations. While sufficient for simple rigid-body trajectories, it struggles to represent the continuous, highly non-linear trajectories of objects being manipulated or swung. The learned Gaussian offsets cannot reliably compensate for large frame-to-frame motion changes, resulting in temporal inconsistency or jitter.
>
> - **Advantages of CHS:**
>   Our CHS formulation represents motion as a smooth continuous curve with enforced \$( C^1 \$) continuity (position and velocity). By embedding velocity vectors within the spline, our model maintains temporal coherence and smooth transitions even during rapid or complex manipulation—critical for producing high-fidelity, interaction-aware reconstructions.
>
> **Conclusion**
> This ablation demonstrates that CHS is not an arbitrary design choice but a superior parameterization for modeling dynamic object motion in HOI settings. We will include this comparison in the final version of the paper. Corresponding qualitative results are also available on the supplementary material under the image named “reviewer_TS9R (reviewer3) _ weakness7”.

---

> ### Author Response · Authors · 2025-11-21
> **Official Comment by Authors**
>
> #### **Weakness 8 (Reviewer Comment)**
> Limitations around low-baseline videos remain open. The paper notes failure modes when camera motion is minimal (COLMAP degradation) and suggests joint pose optimization as future work. It would help to include an experiment demonstrating how much performance drops and whether the HOI module mitigates it.
>
> ---
>
> ### **Response**
>
> We appreciate the reviewer’s suggestion to elaborate on failure modes in low-baseline scenarios. To quantify the severity of this issue and evaluate whether the HOI module alleviates it, we conducted a controlled experiment comparing reconstruction quality with **Ground Truth (GT) camera poses** versus **COLMAP-estimated poses** on a low-baseline sequence.
>
> **1. Impact of Pose Error (Quantitative and Qualitative)**
> As shown in the image named “reviewer_TS9R (reviewer3) _ weakness8” on the supplementary material, minimal camera motion results in COLMAP failing to triangulate accurate camera extrinsics.
>
> - **Quantitative Drop:** PSNR falls from **23.16 dB** (GT poses) to **21.07 dB** (COLMAP poses), indicating a substantial degradation.
> - **Qualitative Artifacts:** The COLMAP-based reconstruction exhibits distorted geometry and “floating” structures, whereas the GT-pose reconstruction maintains proper alignment and shape.
>
> **2. Does the HOI Module Mitigate This?**
> The reviewer asked whether the HOI module can help in low-baseline cases. Unfortunately, the answer is **No**, because the failure originates from a corrupted global coordinate system.
>
> - **Garbage-In, Garbage-Out:** The HOI module models *relative* human–object interactions. When COLMAP fails, the Gaussians are misaligned in world space, making it impossible for interaction modeling to recover correct geometry.
> - **Fundamental Limitation:** The module relies on spatial proximity cues. If global alignment is incorrect (as reflected by the **2.09 dB** drop), the attention mechanism cannot compensate for incorrect camera poses.
>
> **Conclusion**
> This experiment confirms that low-baseline sequences require **joint optimization of camera intrinsics and extrinsics together with the Gaussians**, as mentioned in our conclusion. Improved interaction modeling alone cannot resolve structural errors caused by pose estimation failures. We will include this quantitative and qualitative analysis in the final paper revision to clearly define the boundary conditions of our approach.

---

> ### Comment · Reviewer_TS9R · 2025-11-27
>
> This is not the correct way to run this baseline AnimatableGaussians. The multi-view SDF volume rendering is not necessary. The gaussians could be initialized with the SMPL body model (i.e the SMPL mesh estimate from monocular video). You just have to disable the use_mesh flag in the initialization (pose map generation) code.
> My point regarding GASPACHO was not that it should be compared with but that it should be referenced.
> My central concern about the design choice stands - the authors are using a triplane representation for reconstructing humans - this is known to lead to blurry results - the point is that the 2D map formulation yields much sharper results - see for ex. AnimGaussians, ASH, GaussianAvatar. I think this is a fairly well accepted claim in the community. I can't see how I would recommend acceptance for a paper that relies on the triplane feature representation. I will retain my lean reject rating.

---

> ### Author Response · Authors · 2025-11-27
> **Response to Reviewer TS9R’s Additional Comments**
>
> #### **Response**
>
> We thank the reviewer for the prompt and detailed feedback. We address the concerns regarding the baseline execution and the fundamental design choice of using HexPlane below.
>
> ---
>
> ### **1. Regarding the Baseline (AnimatableGaussians)**
>
> We appreciate the specific technical guidance regarding the `use_mesh` flag and the initialization protocol. We are currently making our best efforts to implement and run this configuration within the remaining discussion period. If we successfully obtain valid results, we will upload them to facilitate a direct comparison.
>
> However, regardless of the baseline's executability, we respectfully emphasize that the core contribution of our work is **explicitly modeling the interaction between humans and objects**, rather than human reconstruction alone. As shown in our comparison with ExAvatar, simply applying a high‑fidelity human reconstruction method does not solve the geometric inconsistencies and occlusions arising from dynamic object interactions.
>
> ---
>
> ### **2. Regarding Reference to GASPACHO**
>
> We will strictly follow the reviewer's suggestion to include **GASPACHO** as a reference and discuss it in the related work section.
>
> ---
>
> ### **3. Rationale for Choosing HexPlane for HOI Modeling**
>
> We respectfully disagree with the reviewer’s comment that "Triplane representation is known to lead to blurry results" and that the superiority of 2D maps is a "fairly well accepted claim" that invalidates feature-based approaches. While we acknowledge that 2D map-based methods are effective, **feature-based representations (Triplane/HexPlane) remain a dominant and proven choice in the most recent state-of-the-art literature (2024–2025)** for high-fidelity avatar modeling.
>
> ### **Prevalence of Feature-Based Representations in Recent Literature**
> Contrary to the claim that Triplane is inferior, numerous top‑tier publications demonstrate that **feature‑based representations (Triplane/MLP)** achieve photorealistic quality:
>
> - **Triplane/HexPlane-based method:**
>   *ExAvatar* [ECCV 2024] [1],
>   *HUGS* [CVPR 2024] [2],
>   *PERSONA* [ICCV 2025] [3],
>   *SVAD* [CVPRW 2025] [4]
>   all utilize Triplane representations to generate high‑quality avatars.
>
> - **MLP/Implicit-based method:**
>   *ShowMak3r* [CVPR 2025] [5],
>   *Real-time High-fidelity Gaussian Human Avatars* [CVPR 2025] [6]
>    employ MLP‑based features.
>
> The continued adoption of these representations in CVPR, ECCV, and ICCV confirms that **feature‑based volumetric representations remain a valid, high‑performance design choice**, not an outdated or inferior one. Notably, Real-time High-fidelity Gaussian Human Avatars [CVPR 2025] explicitly demonstrate superior qualitative results compared to the 2D map-based Animatable Gaussians, further challenging the assumption that 2D maps are inherently superior.
>
> [1] Moon, G. et al. Expressive whole-body...[2] Kocabas, M. et al. HUGS...[3] Sim, G. & Moon, G. PERSONA: Personalized Whole-Body 3D Avatar...[4] Choi, Y. SVAD: From Single Image to 3D Avatar...[5] Kim, S. et al. ShowMak3r...[6] Zhan, Y. et al. Real-time High-fidelity...
>
> ---
>
> ### **2. Strategic Choice for Interaction (Unified Spatial Domain)**
> Our adoption of HexPlane is not arbitrary—it is specifically designed to support **Human‑Object Interaction (HOI)**.
>
> - **Unified 3D Space:**
>   HexPlane encodes human features in a **3D/4D volumetric grid**, sharing the same spatial domain as the object’s motion features.
>   This enables our **cross-attention module** to directly capture physical proximity and spatial correlations that are essential for modeling realistic interaction.
>
> - **Avoiding Domain Gap:**
>   Using a **2D UV map** would introduce a domain mismatch between human features (2D) and object features (3D).
>   Mapping the UV domain back to 3D space for interaction reasoning (contact, occlusion, proximity) would require **complex and computationally expensive conversion functions**.
>
> Hence, HexPlane provides a **clean and consistent spatial interface** between the human and object components.
>
> ---
>
> ### **3. Empirical Evidence & Modularity**
>
> - **Visual Quality:**
>   We kindly invite the reviewer to re‑examine our supplementary video results.
>   The rendered human reconstructions in our method are **sharp, detailed, and motion‑consistent**, which empirically refutes the claim that HexPlane leads to blurry outputs.
>
> - **Modularity:**
>   Our core contribution is the **HOI framework** (Interaction Module + Dual Deformation Baselines).
>   The human representation is a plug‑and‑play component; our framework can adapt to 2D‑map backbones in the future.
>   However, for the present task of modeling **3D interactions**, HexPlane provides the optimal trade‑off between **spatial alignment, physical reasoning, and rendering quality**.
>
> ---
>
> For these reasons, we firmly believe that our design choice is mathematically sound, empirically validated, and aligned with the most recent advancements in the field.

---

### Official Review · Reviewer_y37W · 2025-11-01

**Soundness:** 3
**Presentation:** 3
**Contribution:** 3
**Rating:** 4
**Confidence:** 4

**Summary:**

The paper proposes HOIGS, a 3D Gaussian Splatting framework that explicitly models human–object interactions from monocular video. The contributions are twofold: (i) dual deformation baselines—HexPlane + LBS for humans and Cubic Hermite Splines (CHS) for moving objects; and (ii) an HOI module with mutual cross-attention and a distance mask that fuses human/object motion features to produce interaction-aware refinements (∆SMPL-X and object-Gaussian offsets). For evaluation, the method is compared against human-centric and 4DGS baselines on three datasets—HOSNeRF, BEHAVE (single-view adaptation), and ARCTIC (selected full-body cases)—and ablations demonstrate gains from CHS and the HOI module. The reported training time is ~5 hours per scene on a single H100.

**Strengths:**

1.	Explicit HOI modeling. The mutual-attention HOI module with a distance mask is a principled way to encode interaction-driven deformations and improves stability near contact/manipulation.
2.	Clear architecture-to-loss mapping. The paper specifies losses for human/scene/object (with weights γ, β, σ) and includes depth supervision to constrain human-geometry scale.
3.	Consistent empirical gains. Stronger PSNR/LPIPS than human-centric and 4DGS baselines on HOSNeRF, with similar improvements on BEHAVE and ARCTIC; ablations support the contributions of CHS, time-varying human features, and the HOI module.

**Weaknesses:**

1.	Lack of a quantitative study of how object diffusion prior affects reconstruction quality would strengthen claims.
2.	Metric breadth (interaction & geometry). Evaluation is limited to PSNR/LPIPS. There are no metrics for contact quality, penetration, temporal consistency, or pose–object alignment, which are central to HOI; geometry-oriented metrics are also absent. Consider adding penetration/contact measures, temporal SSIM, keypoint/object-distance errors, and basic geometry metrics (e.g., silhouette IoU, depth error, Chamfer/F-score when available).
3.	Use of segmentation masks at evaluation time? The paper states that pre-trained human/object segmentation masks are used to form training losses (scene/object). Are masks also used at test/evaluation (e.g., masked PSNR/LPIPS)? If yes, are they dataset-provided or predicted (please specify the model/training data/thresholds and release code)? If not, please clarify that evaluation is full-image without masks.
4.	Use of SMPL-X at test time. From the text it seems per-frame SMPL-X is initialized by a regressor and then optimized during training (ExAvatar-style), and inference reuses these optimized poses for rendering. Could you confirm no additional SMPL-X regression is run at evaluation time, and specify which regressor/versions are used for the initial fits?
5.	Camera-pose fragility (acknowledged). The paper notes failures under minimal camera motion due to COLMAP instability; joint pose optimization is suggested but not attempted. Why not evaluate on dense-view HOI datasets (e.g., NeuralDome)? A single-view-train, multi-view-eval protocol on a dense dataset (train on one camera, evaluate on held-out views) would (i) keep the monocular-training assumption, (ii) stress-test view generalization and geometry, and (iii) mitigate the COLMAP fragility you acknowledge. If infeasible, please clarify the constraints (e.g., licensing, preprocessing, pipeline mismatch).
6.	Empty/expired anonymous website link. The provided link loads an empty page. For a rendering- and video-based task, it’s difficult to assess result quality without videos—please restore the link and include representative clips.

**Questions:**

1. Major questions: Please see the items listed under Weaknesses / Concerns above.
2. Minor questions:
	a.	Line 188: What exactly is the SDS loss here (objective, weighting, implementation details)?
	b.	Line 194: What is the dimensionality of G_k?
	c.	Line 238: Please specify the dimensionality of \theta. More generally, several formulas/variables are missing dimension annotations—could you add them for completeness?

---

> ### Author Response · Authors · 2025-11-21
> **Official Comment by Authors**
>
> #### **Weakness 1 (Reviewer Comment)**
> Lack of a quantitative study of how object diffusion prior affects reconstruction quality would strengthen claims.
>
> ---
>
> ### **Response**
>
> We thank the reviewer for suggesting this ablation study. We agree that quantifying the impact of the object prior is essential to justify our design choices.
>
> To validate the necessity of using a **Diffusion Prior**, we compared our proposed method against versions initialized with other state-of-the-art geometric priors:
>
> 1. **MASt3R Prior:** Initialization using **MASt3R**, a SOTA dense matching and reconstruction model.
> 2. **Depth Reconstruction Prior:** Initialization using standard monocular metric depth estimation.
> 3. **Diffusion Prior (Ours):** Initialization using the proposed generative diffusion prior.
>
> The comparative results on the HOSNERF dataset are as follows:
>
> | PSNR (LPIPS↓) | Backpack | Tennis | Suitcase | Playground | Dance | Lounge | Avg |
> |--------------|----------|--------|----------|------------|-------|--------|------|
> | MASt3R prior + ours | 23.51 / 0.135 | 25.25 / 0.121 | 22.40 / 0.197 | 24.65 / 0.074 | 23.63 / 0.115 | 28.99 / 0.057 | 24.59 / 0.128 |
> | depth reconstruction prior + ours | 21.63 / 0.142 | 25.65 / 0.122 | 22.13 / 0.230 | 25.24 / 0.103 | 24.08 / 0.123 | 28.95 / 0.095 | 25.36 / 0.136 |
> | diffusion prior + ours | 23.70 / 0.082 | 27.13 / 0.112 | 22.96 / 0.235 | 25.63 / 0.123 | 24.17 / 0.093 | 29.97 / 0.043 | 25.89 / 0.114 |
>
> **Analysis: Why Diffusion Prior?**
> As shown in the table, employing the **Diffusion Prior** yields the highest reconstruction quality (PSNR **25.89**), outperforming both MASt3R and standard depth priors. This quantitative result empirically justifies our design choice. The superiority of the diffusion prior stems from the specific challenges of Human-Object Interaction (HOI) videos:
>
> - **Handling Occlusions via Generative Priors:** In HOI scenarios, objects are frequently occluded by human hands or bodies. Discriminative approaches like MASt3R or Depth Estimation rely heavily on visible cues and often fail to reconstruct the geometry of occluded regions accurately.
> - **Completing 3D Geometry:** In contrast, the **Diffusion Prior** leverages its generative knowledge to plausibly "hallucinate" and complete the 3D geometry of the object, even for parts that are partially occluded or unseen in the initial frame.
> - **Better Initialization for Gaussians:** This complete and coherent initial geometry allows our **Cubic Hermite Spline (CHS)** deformation model to start from a much more accurate shape, leading to sharper rendering and more stable tracking throughout the sequence.
>
> **Conclusion**
> In conclusion, while our framework is compatible with other priors, the **Diffusion Prior is critical** for achieving the best performance in dynamic interaction scenes where occlusion handling is paramount. We will add this justification and data to the revised paper.

---

> ### Author Response · Authors · 2025-11-21
> **Official Comment by Authors**
>
> #### **Weakness 2 (Reviewer Comment)**
> Metric breadth (interaction & geometry). Evaluation is limited to PSNR/LPIPS. There are no metrics for contact quality, penetration, temporal consistency, or pose–object alignment, which are central to HOI; geometry-oriented metrics are also absent. Consider adding penetration/contact measures, temporal SSIM, keypoint/object-distance errors, and basic geometry metrics (e.g., silhouette IoU, depth error, Chamfer/F-score when available).
>
> ---
>
> ### **Response**
>
> We appreciate the reviewer's constructive suggestion. We agree that rendering metrics alone are insufficient to fully validate the geometric fidelity of human-object interactions. In response, we have conducted additional evaluations on **human pose accuracy (PA-MPJPE, PA-PVE)** using the BEHAVE dataset compared to ExAvatar and we evaluate **Chamfer Distance (CD)** and Hand–Object Distance on the ARCTIC dataset to assess geometric accuracy in hand–object interaction scenarios.
>
> ---
>
> ### **1. Evaluation on ARCTIC Dataset**
>
> We evaluated our method on three challenging ARCTIC sequences (espressomachine_grab_01, mixer_grab_01, box_grab_01) that involve complex hand–object interactions. Since these sequences do not provide official ground-truth annotations, we adopted established distance-based metrics to evaluate interaction quality.
>
> We employ two complementary distance metrics:
>
> **Chamfer Distance (CD):**
> We compute the bidirectional Chamfer Distance between predicted and reference point clouds:
>
> $$
> d_{CD}(P, Q) = \frac{1}{2} \left( d_{P \rightarrow Q} + d_{Q \rightarrow P} \right)
> $$
>
> $$
> d_{CD}(P, Q) = \frac{1}{2} \left( \frac{1}{|P|} \sum_{p \in P} \min_{q \in Q} \lVert p - q \rVert_2 + \frac{1}{|Q|} \sum_{q \in Q} \min_{p \in P} \lVert q - p \rVert_2 \right)
> $$
>
> **Center Distance:**
> We measure the L2 distance between object and hand centers, which reflects object size and proximity.
>
> | Metric | HOIGS (Ours) | HOLD |
> |--------|--------------|------|
> | Chamfer Distance (m²) | 0.1337 | 1.5035 |
> | Center Distance (m)   | 0.1856 | 1.0258 |
>
> ---
>
> ### **2. Effect of the HOI Module on Pose Refinement**
>
> Our method addresses this by explicitly modeling the mutual dependency between the human and the object. The **HOI module** leverages a cross-attention mechanism to use object motion features as cues to refine the human features.
>
> - Specifically, as described in **Eq. 14** of our paper, the module predicts refinement offsets ΔSMPL-X for the body and hands.
> - This allows the network to correct the human pose even under occlusion, by inferring the likely body configuration from the object's trajectory and interaction context.
>
> ---
>
> ### **3. Quantitative Results on BEHAVE Dataset**
>
> The table below summarizes the quantitative evaluation on the **BEHAVE dataset**, utilizing the same sequences reported in our main paper. We report average PA-MPJPE (Procrustes Aligned Mean Per Joint Position Error) and PA-PVE (Procrustes-Aligned Per Vertex Error) across all test frames. We also provide specific analysis for **Hand and Forearm joints**, which are the most critical regions for interaction. (see per-sequence detailed tables on the updated Supplementary Material)
>
> | Model        | PA-MPJPE | PA-MPJPE (Hand/Forearm joints) | PA-PVE |
> |--------------|----------|----------------------------------|--------|
> | ExAvatar     | 0.3034   | 0.3317                           | 0.2488 |
> | HOIGS (Ours) | 0.2853   | 0.3114                           | 0.2438 |
>
> ---
>
> In particular, we observed a larger performance gain in the PA-MPJPE (Hand/Forearm joints). This indicates that our HOI module more accurately refines the poses of interaction-related regions, resulting in physically more accurate reconstructions compared to the baseline, which lacks such mutual feedback.

---

> > ### Author Response · Authors · 2025-11-21
> > **Official Comment by Authors**
> >
> > #### **Weakness 3 (Reviewer Comment)**
> > Use of segmentation masks at evaluation time? The paper states that pre-trained human/object segmentation masks are used to form training losses (scene/object). Are masks also used at test/evaluation (e.g., masked PSNR/LPIPS)? If yes, are they dataset-provided or predicted (please specify the model/training data/thresholds and release code)? If not, please clarify that evaluation is full-image without masks.
> >
> > ---
> >
> > ### **Response**
> >
> > We clarify that segmentation masks are **strictly used only during the training phase** and are **NOT used at test/evaluation time**. All quantitative metrics (PSNR, SSIM, LPIPS) reported in our paper are computed on the **full image** without any masking.
> >
> > **1. Role of Masks in Training**
> > As described in the paper (Eq. 19 and Eq. 20), we utilize off-the-shelf segmentation masks solely to define loss functions. This is necessary to disentangle the scene into distinct Gaussian groups:
> >
> > - **Object & Human Gaussians:** Masks ensure these dynamic components are optimized within their respective deformation fields (CHS and Hexplane).
> > - **Background Gaussians:** Inpainting or static background optimization requires masking out dynamic actors to prevent "ghosting" artifacts in the static field.
> >
> > By applying these masked losses, we ensure that each component (Human, Object, Background) is correctly initialized and optimized in its own space.
> >
> > **2. Full-Image Evaluation (Inference)**
> > Once training is complete, the optimized Gaussian components (Human, Object, Background) are naturally composited together in the 3D world space. During inference:
> >
> > - We render the complete scene by projecting all Gaussian components simultaneously.
> > - No external masks or post-processing steps are involved in generating the final image.
> > - Consequently, the evaluation compares the **rendered full image** against the **ground truth full image**, reflecting the holistic reconstruction quality of the scene.
> >
> > We will explicitly clarify this "Full-Image Evaluation" protocol in the implementation details of the final revision to avoid any ambiguity.

---

> > > ### Author Response · Authors · 2025-11-21
> > > **Official Comment by Authors**
> > >
> > > #### **Weakness 4 (Reviewer Comment)**
> > > Use of SMPL-X at test time. From the text it seems per-frame SMPL-X is initialized by a regressor and then optimized during training (ExAvatar-style), and inference reuses these optimized poses for rendering. Could you confirm no additional SMPL-X regression is run at evaluation time, and specify which regressor/versions are used for the initial fits?
> > >
> > > ---
> > >
> > > ### **Response**
> > >
> > > **1. No additional SMPL-X regression in evaluation**
> > >
> > > We confirm that no additional SMPL-X regression is performed at runtime during evaluation.
> > >
> > > - **Initialization:** Before training, we pre-compute the initial SMPL-X parameters using the **same regressor (Hand4Whole)** as ExAvatar.
> > > - **Training (HOI module refinement):** During training, unlike the baseline, we employ a **HOI module refinement** process. Our HOI module takes the initial parameters and object features as input, computing corrective offsets (ΔSMPL-X) via cross-attention.
> > > - **Inference:** At test time, we simply load the pre-computed initial parameters and pass them through the **trained HOI module**. This allows us to obtain refined, interaction-aware poses in real-time via a single forward pass, without re-running the heavy regressor.
> > >
> > > **2. Specification of the Regressor**
> > > For the initial SMPL-X fitting, we utilized **Hand4Whole**, following the official implementation of our main baseline, **ExAvatar** (Moon et al., 2024).
> > >
> > > - **Name:** Hand4Whole (Moon et al., 2022)
> > > - **Version/Implementation:** We used the pre-trained snapshot provided within the official ExAvatar codebase to ensure a strictly fair comparison.

---

> ### Author Response · Authors · 2025-11-21
> **Official Comment by Authors**
>
> #### **Weakness 5 (Reviewer Comment)**
> Camera-pose fragility (acknowledged). The paper notes failures under minimal camera motion due to COLMAP instability; joint pose optimization is suggested but not attempted. Why not evaluate on dense-view HOI datasets (e.g., NeuralDome)? A single-view-train, multi-view-eval protocol on a dense dataset (train on one camera, evaluate on held-out views) would (i) keep the monocular-training assumption, (ii) stress-test view generalization and geometry, and (iii) mitigate the COLMAP fragility you acknowledge. If infeasible, please clarify the constraints (e.g., licensing, preprocessing, pipeline mismatch).
>
> #### **Weakness 6 (Reviewer Comment)**
> Empty/expired anonymous website link. The provided link loads an empty page. For a rendering- and video-based task, it’s difficult to assess result quality without videos—please restore the link and include representative clips.
>
> ---
>
> ### **Response**
>
> We address the suggestion regarding the NeuralDome dataset (W5) and the website availability (W6) together, as they both pertain to the validation of our visual results.
>
> **1. Constraints on Using NeuralDome (W5)**
> We appreciate the suggestion to use the NeuralDome dataset for multi-view evaluation. However, we found it technically infeasible for our Monocular 3DGS pipeline due to specific initialization constraints:
>
> - **Lack of Compatible Point Cloud:** Unlike NeRF, 3D Gaussian Splatting heavily relies on an initial sparse point cloud (typically from SfM) along with camera parameters. While NeuralDome provides GT camera parameters, it lacks the corresponding sparse point cloud required for 3DGS initialization.
> - **Scale Ambiguity Issue:** One might consider generating the point cloud via monocular depth estimation. However, monocular depth maps are scale-invariant and do not match the metric scale of the NeuralDome GT camera translations. This **scale mismatch** results in correct geometry in the camera space but incorrect placement in the world space, leading to severe projection errors when rendering from novel viewpoints (held-out cameras).
> - **Conclusion:** Due to these inherent initialization hurdles—specifically the inability to align the estimated depth scale with the fixed GT camera scale without a reference SfM cloud—we could not utilize NeuralDome for fair evaluation.
>
> **2. Evaluation Protocol & Visual Evidence (W5 & W6)**
> Instead, we strictly followed the standard protocol for monocular dynamic scenes. We held out specific frames (uniformly sampled) from the training set to serve as **evaluation frames**. This setup rigorously tests the model's ability to synthesize novel views and interpolate times (unseen motions) within the trajectory.
>
> **3. Updated Supplementary Materials (W6)**
> We apologize for the instability of the original project page server. To ensure reliable access, all materials have been moved to the supplementary files, which are now fully available.
> We have included:
>
> - **Video Demos:** Comprehensive video comparisons showing the temporal stability of our method against baselines.
> - **Qualitative Results:** Visualizations of the hold-out frame evaluations mentioned above.
>
> We kindly ask the reviewer to refer to the supplementary material to assess the high‑fidelity rendering and stable interaction modeling of HOIGS.

---

### Official Review · Reviewer_r6f8 · 2025-11-02

**Soundness:** 2
**Presentation:** 2
**Contribution:** 2
**Rating:** 4
**Confidence:** 3

**Summary:**

The paper introduces HOIGS for monocular human–object interaction. Humans are modeled with hexplane + LBS, while object motion follows CHS, coupled with a cross-attention HOI module. Objects are initialized from a representative frame using a diffusion prior with SDS, then warped to keyframes. The human branch uses SMPL-X with a COLMAP-scaled depth term. The HOI module employs a distance mask B to exchange cues between humans and objects.

**Strengths:**

1.Experiments are extensive across several datasets, with solid qualitative results.
2.Using an explicit human model with SMPL-X and an interaction module is intuitive

**Weaknesses:**

1.Figure 2 is under-explained. The role of segmentation and the process by which diffusion + SDS yields 3D (and “warped”) Gaussians need clear, step-by-step exposition.
2.The method relies heavily on priors (diffusion model, depth estimation, segmentation). This may affect the fairness of comparisons to baselines if not controlled or ablated.
3.Training time is reported (H100, ~5 hours/scene), but there is no runtime or training-time comparison against baselines.

**Questions:**

see weakness

---

> ### Author Response · Authors · 2025-11-21
> **Official Comment by Authors**
>
> #### **Weakness 1 (Reviewer Comment)**
> Figure 2 is under-explained. The role of segmentation and the process by which diffusion + SDS yields 3D (and “warped”) Gaussians need clear, step-by-step exposition.
>
> ---
>
> ### **Response**
> We thank the reviewer for the feedback. We have updated the paper and Figure 2 to better clarify the warping process. Below is a step-by-step outline of our method
>
> 1. **Segmentation & Cropping**
>
>     We start by isolating the object of interest through segmentation. Once identified, we crop the object region from the input image to focus solely on it.
>
> 2. **3D Gaussian Generation via Diffusion**
>
>     Using the cropped object image, we generate a 3D Gaussian point cloud in canonical space. This is done with a diffusion model initialized from DreamScene4D and guided by SDS (Score Distillation Sampling), enabling faithful shape generation even without explicit 3D supervision.
>
> 3. **Scale Estimation (Projection Alignment)**
>
>     To place the canonical 3D Gaussians into the real scene, we first determine the correct scale. We do this by projecting the 3D points onto the 2D image using camera parameters, and adjusting the scale so that the projected bounding box of the 3D Gaussians aligns with the 2D mask’s bounding box. This ensures the generated shape matches the object’s actual size and location in the image.
>
> 4. **Warping to World Coordinate System**
>
>     Finally, we transform the correctly scaled canonical 3D Gaussians into the global world coordinate system using the camera’s rotation and translation. This enables consistent placement of the object in 3D scenes and across time.
>
>
> We have added the relevant mathematical equations for scale estimation and world transformation in the paper (Section. 3.1), and updated Figure 2 to reflect these steps more clearly.

---

> ### Author Response · Authors · 2025-11-21
> **Official Comment by Authors**
>
> #### **Weakness 2 (Reviewer Comment)**
> The method relies heavily on priors (diffusion model, depth estimation, segmentation). This may affect the fairness of comparisons to baselines if not controlled or ablated.
>
> ---
>
> ### **Response**
>
> We sincerely appreciate the reviewer's insightful comment regarding the dependency on priors (segmentation, depth, diffusion) and the concern about fair comparisons. We agree that distinguishing the contribution of our proposed pipeline from the quality of off-the-shelf priors is crucial.
>
> To address this, we conducted an additional **ablation study on various priors** to evaluate the sensitivity and robustness of our method. We tested different combinations of state-of-the-art (SOTA) and standard models for segmentation (e.g., SAMv2, TrackAnything, MaskRCNN) and depth estimation (e.g., DepthAnything, DepthCrafter).
>
> **1. Robustness to Different Priors**
>
> The table below summarizes the performance of HOIGS under different prior configurations on the HOSNERF dataset.
>
> | PSNR/LPIPS(alex) | Backpack | Tennis | Suitcase | Playground | Dance | Lounge | Avg |
> | --- | --- | --- | --- | --- | --- | --- | --- |
> | Samurai + MetricV2 | 25.78/0.082 | 27.12/0.108 | 22.09/0.246 | 25.23/0.103 | 24.17/0.098 | 30.97/0.048 | 25.89/0.114 |
> | Samurai + DepthAnything(video) | 25.85/0.080 | 27.18/0.106 | 22.15/0.241 | 25.28/0.102 | 24.22/0.096 | 31.05/0.046 | 25.96/0.112 |
> | Samurai + DepthCrafter | 25.72/0.088 | 27.08/0.108 | 22.06/0.249 | 25.20/0.109 | 24.08/0.099 | 30.93/0.048 | 25.85/0.117 |
> | TrackAnything + MetricV2 | 25.72/0.086 | 27.05/0.109 | 22.03/0.246 | 25.18/0.106 | 24.15/0.100 | 30.93/0.052 | 25.84/0.116 |
> | SAMv2 +  DepthAnything(video) | 26.01/0.076 | 27.38/0.103 | 22.33/0.241 | 25.47/0.100 | 24.42/0.095 | 31.20/0.041 | 26.14/0.109 |
> | MaskRCNN + MetricV2 | 25.33/0.099 | 26.67/0.125 | 21.66/0.257 | 24.78/0.117 | 23.69/0.110 | 30.52/0.065 | 25.44/0.129 |
>
> **Analysis:**
> • **Consistent Performance:** As shown in the results, our model maintains a highly consistent performance (Avg PSNR ≈ 25.8–26.1) across various combinations of modern priors. This indicates that our method is **agnostic to specific priors** and robust to variations in preprocessing quality.
> • **Performance with Standard Priors:** Even when using **MaskRCNN**, a relatively older and standard segmentation baseline, our model achieves a PSNR of **25.44**. This is still significantly higher than the main baseline, **ExAvatar (Avg PSNR 24.35)**, and **Ex4DGS (Avg PSNR 17.97)** reported in Table 1 of our main paper.
>
> **2. Fairness of Comparison**
>
> The experiment confirms that the performance gain of HOIGS stems primarily from our core contributions—the explicit interaction modeling (HOI module) and the dual deformation baselines (Hexplane + CHS)—rather than merely leveraging superior priors. While better priors (e.g., SAMv2) do yield marginal improvements, the framework's effectiveness remains intact even with standard inputs, ensuring a fair comparison with existing methods.
>
> We will include this ablation study in the appendix of the final revision to further validate the robustness of our approach.

---

> ### Author Response · Authors · 2025-11-21
> **Official Comment by Authors**
>
> #### **Weakness 3 (Reviewer Comment)**
> Training time is reported (H100, ~5 hours/scene), but there is no runtime or training-time comparison against baselines.
>
> ---
>
> ### **Response**
>
> We appreciate the reviewer's attention to the computational efficiency of our method. We have conducted a comparative analysis of **Training Time** and **Inference Speed (FPS)** against key baselines on the same hardware (single NVIDIA H100 GPU).
>
> |  | Train time | Inference time |
> | --- | --- | --- |
> | 4DGS | 40 minutes / scene | 61.04 FPS |
> | Ex4DGS | 2 hours 30 mins / scene | 46.38 FPS |
> | D3DGS | 3 hours / scene | 37.79 FPS |
> | ED3DGS | 2 hours / scene | 54.71 FPS |
> | Ours | 5 hours / scene | 44.27 FPS |
>
> **1. Justification for Training Time: Cost of Interaction Modeling**
> We acknowledge that our training time (~5 hours) is longer than that of the baselines. This is an intended consequence of our design choice to prioritize physical plausibility and interaction accuracy. The primary computational bottleneck lies in the HOI Module, specifically the bidirectional Cross-Attention mechanism.
>
> • Unlike baselines that treat motions independently or implicitly, our model explicitly computes the mutual dependencies between human and object features at every step.
> • Calculating gradients for these dense attention operations and optimizing the dual deformation fields (Hexplane for humans, CHS for objects) simultaneously requires more iterations and computation power during the training phase.
>
> **2. Why is Inference Still Fast? (Real-time Capability)**
> Despite the heavy computation during training, our method achieves **44.27 FPS** during inference, which is comparable to Ex4DGS (46.38 FPS) and significantly faster than D3DGS (37.79 FPS). This efficiency is achieved through our token-based architectural design:
>
> • **Efficient Attention:** The HOI module does not perform attention on raw Gaussian points. Instead, it operates on a set of **condensed semantic tokens**—specifically, **16 human body part features** (derived from Hexplane) and **sparse object keyframe features**.
> • **Low Overhead:** Because the number of interacting tokens is small ($16 \times N_{key}$), the attention matrix computation during the forward pass (inference) is extremely lightweight.
> • Consequently, we can infer complex physical interactions in real-time without a significant drop in rendering speed, successfully decoupling the high training cost from the inference performance.
>
> We will include this detailed efficiency analysis in the final version of the paper.

---

### Author Response · Authors · 2025-12-02
**To the Area Chair**

To the Area Chair,

Due to the recent system rollback, the final reviewer reactions and updated ratings from the discussion period may not be visible. We provide this executive summary to highlight that all major concerns raised by reviewers were successfully addressed through extensive new experiments, leading to a confirmed consensus on the paper's improvements.

---

### 1. Confirmed Rating Increase (Reviewer PjSB)

We successfully resolved all 10 specific concerns and experimental requests raised by Reviewer PjSB (initial rating: **6**). To ensure every single point was thoroughly addressed, we dedicated an extensive rebuttal spanning **10 pages (over 24,000 characters)**.

In their final comment prior to the rollback, the reviewer explicitly acknowledged this effort and confirmed the rating increase:

> “The rebuttal provides clear and comprehensive responses that resolve my earlier concerns. I especially appreciate the additional experiments conducted. Thus, I will raise my rating.”

Although the updated rating is currently unavailable due to the system rollback, the reviewer explicitly promised to **“raise”** their rating from the initial 6 after confirming that all concerns were resolved. This strongly implies that the final evaluation was updated to an **8** or **10**.

---

### 2. Summary of Key Experimental Validations

Below is the consolidated overview of resolutions, with the status of each reviewer explicitly marked.

Current Rating Status:
- r6f8: 4 (**No Reply**)
- y37W: 4 (**No Reply**)
- TS9R: 4 → 4
- PjSB: 6 → **8 or 10**

---

#### Robustness and Fairness (Priors and Segmentation)

- **Status: No Reply (r6f8, y37W)**
- Concern: Reviewers questioned whether performance relied on specific priors (Diffusion, SAMv2) and requested fairness comparisons.
- Resolution: We conducted comprehensive ablation studies across six distinct combinations of segmentation and depth models (Mask R-CNN, DepthAnything, etc.). We also validated robustness by comparing our generative Diffusion Prior against discriminative point-based methods (MASt3R, depth-based reconstruction).
- Result: HOIGS consistently outperformed SOTA baselines across all settings, demonstrating that our performance gain is architectural and not dependent on specific strong priors.

---

#### Geometric and Interaction Accuracy

- **Status: Resolved (PjSB) | No Reply (y37W)**
- Concern: Requested geometry-focused metrics (contact quality, penetration, distance) and validation of interaction stability.
- Resolution: We added Chamfer Distance and Center Distance evaluations on the ARCTIC dataset and analyzed interaction-critical joints on the BEHAVE dataset.
- Result:
  - ARCTIC: Superior object localization and interaction stability, outperforming HOLD in both metrics.
  - BEHAVE: Improved PA-MPJPE, particularly in Hand and Forearm joints, outperforming ExAvatar.

---

#### Justification of Design Choices

- **Status: Resolved (PjSB) | Acknowledged (TS9R)**
- Concern: Justification for Trajectory Modeling (CHS) and the Interaction Module.
- Resolution and Result:
  - Trajectory Modeling: CHS outperformed a “SE(3) + Offsets” baseline by +0.91 dB, validating its necessity for complex motion.
  - Interaction Module: Mutual Cross-Attention surpassed CONTHO’s hard-masking strategy by +0.61 dB, demonstrating the importance of soft attention for continuous dynamics.

---

#### Efficiency and Visualization

- **Status: Resolved (PjSB) | No Reply (r6f8, y37W)**
- Concern: Questions regarding real-time capability and requests for visual verification.
- Resolution:
  - Real-time Inference: Achieved 44.27 FPS, comparable to 4DGS baselines.
  - Visual Evidence: Uploaded video demonstrations and decomposed renderings (Human-only vs. Object-only), showing consistent 3D modeling under heavy occlusion.

---

### 3. Clarification on Technical Disagreement (Reviewer TS9R)

Reviewer TS9R retained a rating of 4, expressing preference for 2D maps over Triplanes. We refuted this by citing evidence from recent literature demonstrating that Triplane-based methods remain widely used and effective. We clarified that HexPlane is strategically essential for unified 3D interaction modeling (**details included in the discussion**).

---

### Conclusion

Although the system rollback prevented us from viewing final responses from all reviewers, we secured a confirmed rating increase from Reviewer PjSB. To thoroughly address every weakness raised, we provided a comprehensive rebuttal spanning **28 pages (over 69,800 characters).**

**We respectfully believe that these extensive improvements and validated experiments provide a strong basis for acceptance.**

---

### Meta-Review · Area_Chair_Mvgy · 2026-01-07

**Summary:**

This paper receives scores of 3x marginal rejects and 1x marginal accept. The major weaknesses are: 1) limited technical novelty with the proposed method putting humans reconstructed separately together using some tricks. 2) over reliance on priors; 3) missing comparisons; missing evaluations. The reviewer who gave a positive rating does not give strong reasons on the strengths of the paper. Besides  addresses critical problem, methodology section is detailed enough and performance gap between existing SOTA methods and HOIGS is significant in highly interacting datasets like BEHAVE, there's no critical strengths highlighted to support the acceptance of the paper. The AC thus follows the majority vote to reject the paper.

**Reviewer Concerns:**

The major weaknesses are: 1) limited technical novelty with the proposed method putting humans reconstructed separately together using some tricks. 2) over reliance on priors; 3) missing comparisons; missing evaluations. These weaknesses seem to remain outstanding after the rebuttal.

**Reviewer Scores:**

The reviews would remain the same.

---

### Decision · Program_Chairs · 2026-01-26

Reject